# ATP biphasically modulates LLPS of TDP-43 PLD by specifically binding arginine residues

Mei Dang[1], Liangzhong Lim[1], Jian Kang [1] & Jianxing Song [1✉]

Mysteriously neurons maintain ATP concentrations of ~3 mM but whether ATP modulates TDP-43 LLPS remains completely unexplored. Here we characterized the effect of ATP on LLPS of TDP-43 PLD and seven mutants by DIC and NMR. The results revealed: 1) ATP induces and subsequently dissolves LLPS of TDP-43 PLD by specifically binding Arg saturated at 1:100. 2) ATP modifies the conformation-specific electrostatic property beyond just imposing screening effect. 3) Reversibility of LLPS of TDP-43 PLD and further exaggeration into aggregation appear to be controlled by a delicate network composed of both attractive and inhibitory interactions. Results together establish that ATP might be a universal but specific regulator for most, if not all, R-containing intrinsically-disordered regions by altering physicochemical properties, conformations, dynamics, LLPS and aggregation. Under physiological conditions, TDP-43 is highly bound with ATP and thus inhibited for LLPS, highlighting a central role of ATP in cell physiology, pathology and aging.

[1] Department of Biological Sciences, Faculty of Science, National University of Singapore, Singapore, Singapore. ✉email: dbssjx@nus.edu.sg

Aggregation of TAR DNA-binding protein 43 (TDP-43) in the cytoplasm of motor neurons is a common pathological hallmark of most cases of amyotrophic lateral sclerosis (ALS), which has also been found in other major neurodegenerative diseases, including Alzheimer's (AD), Parkinson's (PD), frontotemporal dementia (FTD), and Huntington's (HD) diseases[1–3]. 414-residue TDP-43 consists of the folded N-terminal[4–6] and two RNA recognition motif (RRM) domains[7], as well as C-terminal low-complexity (LC) domain over the residues 267–414 which is intrinsically disordered with the amino acid composition similar to those of yeast prion proteins driving the formation of infectious conformations, thus designated as the prion-like domain (PLD)[8–18]. Surprisingly, despite being intrinsically disordered, TDP-43 PLD hosts almost all ALS-causing mutations and drives liquid–liquid phase separation (LLPS) of TDP-43, which is essential for functionally forming stress granules (SGs) composed of protein and RNA components but can also exaggerate into pathological aggregation or amyloid fibrils[17].

Very recently, a neuronal cell death mechanism has been decrypted which is initiated by LLPS of cytoplasmic TDP-43 independent of forming conventional SGs[2]. Briefly, upon a pathological accumulation in the cytoplasm, TDP-43 undergoes LLPS to form liquid droplets without classical stress granule (SG) markers TIA1 and G3BP1. Furthermore, cytoplasmic TDP-43 droplets act to recruit importin-α and Nup62 as well as induce mislocalization of RanGap1, Ran, and Nup107, consequently resulting in inhibition of nucleocytoplasmic transport, clearance of nuclear TDP-43, and finally cell death. This indicates that under pathological conditions, TDP-43 per se undergoes LLPS to initiate ALS, which is independent of the functional LLPS needing RNA and other proteins to form SGs[2,17].

Mysteriously, all living cells maintain ATP concentrations of 2–12 mM, which are much higher than those required for its previously known functions at micromolar concentrations[19–21]. Only recently, it was decoded that ATP at concentrations >5 mM hydrotropically dissolves LLPS of RNA-binding proteins as well as prevents/inhibits protein aggregation/fibrillation[20,21]. We further found that by specific bivalent binding, ATP also mediates LLPS of the intrinsically disordered RGG-rich domain[22] and inhibits fibrillation of the folded RRM domain[23] of FUS. Intriguingly, we found that the capacity of ATP in modulating LLPS of FUS domains is highly dependent on its triphosphate chain, which was experimentally shown to own unique hydration properties:[24] in addition to the "constrained water" with motions slower than those of bulk water, triphosphate chain also has the "hypermobile water" with motions even faster than those of bulk water (Supplementary Fig. 1a). Noticeably, despite having the lowest ATP concentrations among all human cell types, neurons still maintain ATP concentrations of ~3 mM[19]. So two key questions of both physiological and pathological significance arise: (1) can ATP modulate LLPS of TDP-43 PLD that is not RGG-/R-rich? (2) If yes, what is the underlying molecular mechanism?

So far, due to the challenge in characterizing LLPS of the intrinsically aggregation-prone TDP-43 PLD, the exact driving forces for its LLPS are still under debate despite extensive studies[11–15]. In particular, it remains completely unexplored for the effect and mechanism of ATP on LLPS of TDP-43 PLD. Here by DIC microscopy and NMR spectroscopy, we aimed to address the two questions by characterizing the effect of ATP on LLPS as well as its molecular interactions with the wild-type (WT) and seven TDP-43 PLD mutants. We found that ATP is indeed capable of inducing at low concentrations but dissolving LLPS of TDP-43 PLD at high concentrations. Unambiguous NMR assignments together with mutagenesis studies reveal that although 150-residue TDP-43 PLD only contains five Arg

residues, ATP achieves the biphasic modulation of its LLPS by specifically binding to Arg residues. Further studies on seven mutated/deleted variants decode that LLPS of TDP-43 PLD and its reversibility are in fact controlled by a delicate network which is constituted not only by the attractive interactions as previously identified[11–14] but also by the inhibitory interaction associated with Arg residues. Our results altogether establish that ATP can specifically bind Arg residues within the intrinsically disordered regions (IDRs) which do not need to be RGG-/RG- or R-rich as the FUS RGG-rich domain[22]. These results support the emerging notion that ATP with concentrations >mM has a category of previously unknown functions operating to control protein homeostasis with diverse mechanisms that are fundamentally critical for cell physiology, pathology, and aging.

## Results

**ATP biphasically modulates LLPS of TDP-43 PLD**. TDP-43 has a long C-terminal PLD from residues 267–414 containing six basic residues: Arg268, Arg272, Arg275, Arg293, and Arg361, as well as Lys408 (Fig. 1a). Previous NMR studies from others and we indicate that TDP-43 PLD is intrinsically disordered but contains a unique central hydrophobic region over residues 311–343 adopting a partially folded helical conformation[11,12,16,25]. Based on our previous screening of protein concentrations and buffer conditions for studying the amyloid formation[16] and LLPS[26] of TDP-43 PLD, here we further screened and found that at protein concentrations of 15 µM or lower, no LLPS or aggregation was detected for TDP-43 PLD without ATP in 10 mM sodium phosphate or acetate buffers from pH 4 to 8 at 25 °C. On the other hand, at high concentrations such as >50 µM, it could phase separate without ATP but would become precipitated rapidly as exemplified in Supplementary Fig. 1b, which was extensively observed at high TDP-43 PLD concentrations[11,12,16,26].

Nevertheless, upon titration with ATP, TDP-43 PLD at 15 µM was induced to phase separate, which reached the highest turbidity (absorption at 600 nm) values (1.62 and 0.94, respectively at pH 7.0 and 5.5) at a molar ratio of 1:100 (PLD:ATP) (Fig. 1b and Supplementary Fig. 1c). Further increase of ATP concentrations led to the reduction of turbidity and at 1:1500 the turbidity values became only ~0.05 at both pH. DIC visualization showed that without ATP, no droplets or aggregates were detected at both pH 7.0 and 5.5. Nevertheless, upon adding ATP, the dynamic droplets were formed at 1:25 with the maximal diameter of ~1 µm and at 1:100, many droplets were formed (Fig. 1c). Further increase of ATP concentrations led to reduction of the droplet number. At 1:750 only very few droplets could be detected, while at 1:1000, the droplets were completely dissolved. The overall pattern of the induction and dissolution at pH 7.0 is the same as at pH 5.5, except that the number of the droplets was higher than that at pH 5.5 while the sizes showed no large difference (Fig. 1c), thus resulting in the higher turbidity (Fig. 1b).

We have also titrated a WT sample at 7.5 µM with ATP. The overall pattern is highly similar to those of the samples at 15 µM. Briefly, the turbidity reached the highest at 1:100 but the value is only 0.43 (Fig. 1b). DIC characterization showed that even at 1:100, the number of the droplets is less than that at 15 µM. Interestingly, at 1:400, the turbidity reduced to 0.1 and droplets became undetectable (Fig. 1d).

**ATP achieves the biphasic modulation of LLPS by specifically binding**. To probe the molecular interactions underlying ATP-induced induction and dissolution of LLPS, we conducted ATP titrations on TDP-43 PLD at 15 µM in the same buffer at pH 5.5 as monitored by NMR HSQC spectroscopy, which is particularly

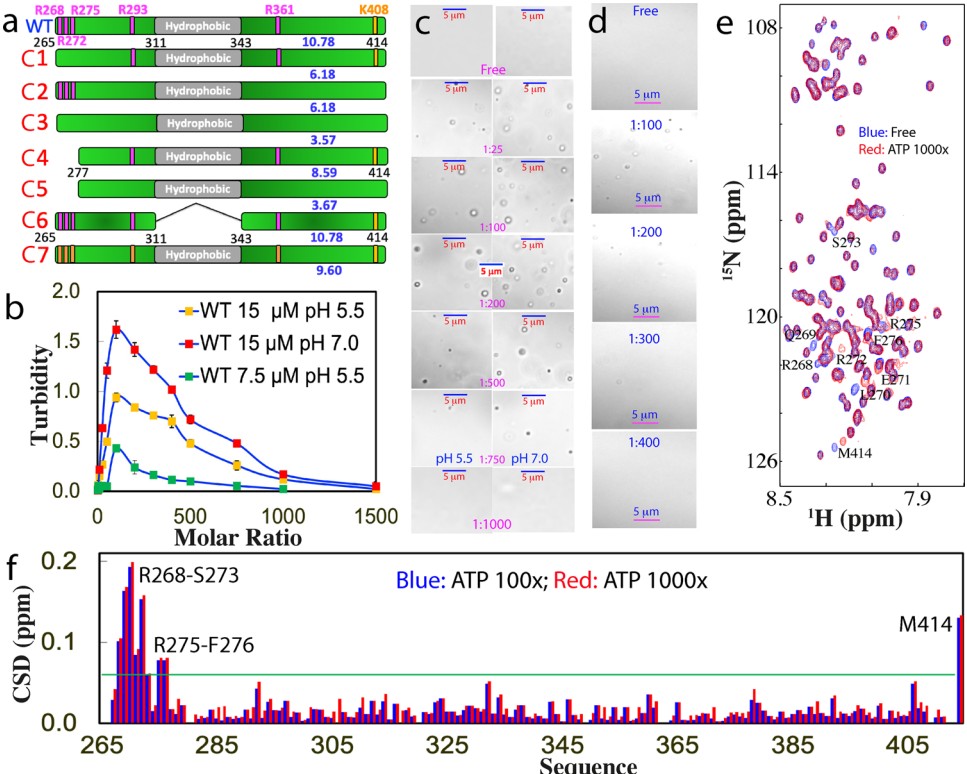

**Fig. 1 DIC and NMR visualization of ATP in modulating LLPS of TDP-43 PLD. a** Schematic representation of WT and seven mutants of TDP-43 PLD with their pI values indicated. **b** Turbidity (absorption at 600 nm) of TDP-43 PLD at 15 μM in the presence of ATP at different molar ratios in 10 mM sodium phosphate buffer at pH 5.5 and 7.0, as well as at 7.5 μM at pH 5.5. **c** DIC images of TDP-43 PLD at 15 μM in the presence of ATP at different molar ratios in the same buffer respective at pH 5.5 and 7.0. **d** DIC images of TDP-43 PLD at 7.5 μM in the presence of ATP at different molar ratios at pH 5.5. **e** $^1$H-$^{15}$N NMR HSQC spectra of the $^{15}$N-labeled TDP-43 PLD at 15 μM in the absence (blue) and in the presence of ATP at a molar ratio of 1:1000 (red) in 10 mM sodium phosphate buffer (pH 5.5). **f** Chemical shift difference (CSD) of TDP-43 PLD between the free state and in the presence ATP at 1:100 (blue) and 1:1000 (red), respectively. The green line is used to indicate the value: average + STD at the ATP ratio of 1:1000. The residues with CSD values >average + SD are defined as significantly perturbed residues.

powerful in detecting very weak perturbations by ATP at residue-specific resolution[22,23]. We selected pH 5.5 for all NMR studies because: (1) several mutants showed severe aggregation above pH 6.0; (2) many HSQC peaks disappeared at pH 7.0 due to the enhanced exchange of amide protons with bulk water and/or dynamic self-association as previously observed[11,12,16,22].

Very unexpectedly, although ATP is a hydrotropic molecule that contains both hydrophobic purine ring and highly negatively charged triphosphate chain (Supplementary Fig. 1a), it only induces the shift of a small set of HSQC peaks upon addition even up to 1:1000 (Fig. 1e). Interestingly, HSQC peaks of TDP-43 PLD in the presence of ATP at 1:1000 appeared to be broader than those in the free state, which may be due to the alternation of dynamics, or/and association of PLD upon binding to ATP. Detailed analysis revealed that except for the last residue Met414, the residues with significant shifts are clustered over N-terminal three Arg residues, which include Arg268-Gln269-L270-Glu271-Arg272-Ser273 and Arg275-Phe276. As it is well-documented that the C-terminal residue of almost all proteins with a free carboxyl group is very sensitive to minor changes of buffer conditions such as salt concentrations and pH, the addition of the highly charged ATP is expected to largely perturb its chemical shift. Noticeably. the shifting process of these residues is largely saturated at the molar ratio of 1:100 (PLD:ATP) (Fig. 1f).

We subsequently titrated TDP-43 PLD with AMP under the same experimental conditions. No large increase in turbidity was observed even with AMP/PLD ratio reaching up to 1:1500 (Supplementary Fig. 2a). DIC visualization indicated that no

droplets or aggregates were detectable at all points of AMP concentrations. We also probed the effect of AMP by NMR and the results showed that AMP induced no large shifts of HSQC peaks of TDP-43 PLD even up to 1:500 (Supplementary Fig. 2b). Noticeably, further increase of AMP concentrations triggered the reduction of peak intensity and at 1:1000, HSQC signals were too weak to be detectable. This observation implies that AMP might induce the dynamic association of TDP-43 PLD which led to the disappearance of HSQC peaks but was undetectable by turbidity and DIC.

We further titrated TDP-43 PLD under the same conditions as ADP (Supplementary Figs. 2a and 3). Upon adding ADP, the turbidity of the sample increased and at 1:1500, the turbidity reached 1.31, which is even higher than the largest value titrated by ATP (Fig. 1b). Unexpectedly, however, DIC visualization showed that at 1:400, ADP induced the formation of detectable aggregates which became larger upon increasing ADP concentrations. At 1:1500, very large aggregates >5 μm were formed which became visible precipitates after 10 min (Supplementary Fig. 3a). Furthermore, the NMR characterization showed that even at 1:25, the HSQC peaks of the residues around the N-terminal three Arg and last residues underwent obvious shifts, whose pattern is highly similar to that induced by ATP (Supplementary Fig. 3b). However, upon further adding ADP, the HSQC signals became too weak to be detected.

We also set to assess whether the inducing effect of ATP is simply due to the electrostatic screening effect by titrating NaCl into TDP-43 PLD. The results showed a very minor increase of

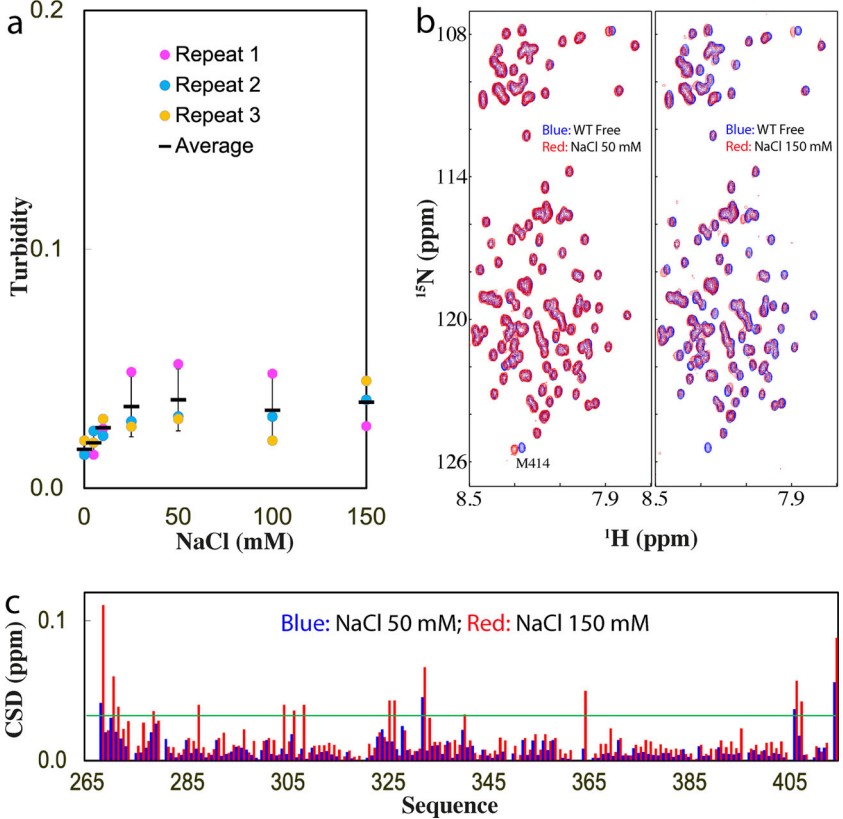

**Fig. 2 Characterization of the interaction of TDP-43 PLD with NaCl. a** Turbidity of TDP-43 PLD (three measurements) at 15 μM in 10 mM sodium phosphate buffer (pH 5.5) in the presence of NaCl at different concentrations. **b** HSQC spectra of the $^{15}$N-labeled TDP-43 PLD at 15 μM in the same buffer at pH 5.5 in the free state (blue) and in the presence of NaCl (red) at 50 and 150 mM, respectively. **c** Chemical shift difference (CSD) of TDP-43 PLD between the free state and in the presence of NaCl at 50 mM (blue) and 150 mM (red). The green line is used to indicate the value: average + SD at 150 mM NaCl.

turbidity (Fig. 2a) as well as no formation of droplets and aggregates as visualized by DIC with NaCl concentrations even up to 150 mM. On the other hand, as monitored by NMR, at 50 mM of NaCl, only HSQC peak of Met414 shifted, while at 150 mM, a set of other HSQC peaks further shifted (Fig. 2b, c), whose pattern, however, is different from that induced by ATP (Fig. 1e). The results indicate that at 15 μM and pH 5.5, NaCl has no detectable capacity to induce LLPS of TDP-43 PLD even up to 150 mM. As judged from the results with ATP, ADP, AMP, and NaCl, ATP appears to achieve the modulation of LLPS by more than a simple electrostatic effect, which is in fact dependent on the presence of the triphosphate chain, as we previously observed on FUS domains[22].

**The role of Arg residues in LLPS of TDP-43 PLD**. As NMR titrations showed that other residues clustered over N-terminal three Arg residues are also largely perturbed, it is hence essential to determine whether their shift is resulting from the direct binding of these residues with ATP, or due to the secondary perturbation by the binding of ATP to the neighboring three Arg residues. Therefore, we constructed three mutants of TDP-43 PLD by site-directed mutagenesis, namely C1 with Arg268, Arg272, and Arg275 mutated to Ala, C2 with Arg293, Arg361, and Lys408 mutated to Ala, as well as C3 with all Arg and Lys mutated to Ala (Fig. 1a).

First, we have assessed the capacity of three mutants in phase separation at different protein concentrations, buffer pH, and NaCl concentrations. We found that all three mutants became severely prone to aggregation at pH above 6. While the proneness

of C1 and C2 to aggregation above pH 6.0 might be explained by their pI of ~6.2, it is unexpected for the proneness of C3 with pI of ~3.6 (Fig. 1a), thus implying that factors other than the simple electrostatic effect also contribute to the aggregation of TDP-43 PLD as we previously observed[16]. At pH 5.5, the C1 and C2 samples showed no severe aggregation at 15 μM in 10 mM phosphate buffer. The C3 sample at 15 μM started to aggregate at pH 5.5 30 min after the sample preparation even with aggregates removed by centrifuge. In particular, the addition of NaCl even at 50 mM induced visible precipitation for all three samples. Nevertheless, we have successfully acquired HSQC spectra at 15 μM in 10 mM phosphate buffer (pH 5.5) for the freshly prepared samples of three constructs. All three mutants have HSQC spectra with the majority of peaks superimposable to those of WT (Supplementary Fig. 4), indicating no global alternation of the solution conformations of three mutants.

Subsequently, we titrated ATP into the samples of the three mutants at 15 μM in the same buffer. As shown in Fig. 3a, upon adding ATP, the turbidity of the C1 sample increased and reached the highest (0.78) at 1:200; and subsequently reduced. On the other hand, DIC visualization showed that without ATP, the C1 sample had no droplets or aggregates. With the addition of ATP, the droplets were formed at 1:25 but even at 1:200 with the highest turbidity, the droplet number is much less than what was observed on WT (Fig. 3b). The droplets also started to be dissolved with further addition of ATP and at 1:1000, the droplet became completely dissolved.

In the absence of ATP, the C2 sample also had no large turbidity reading. Upon adding ATP, the turbidity increased

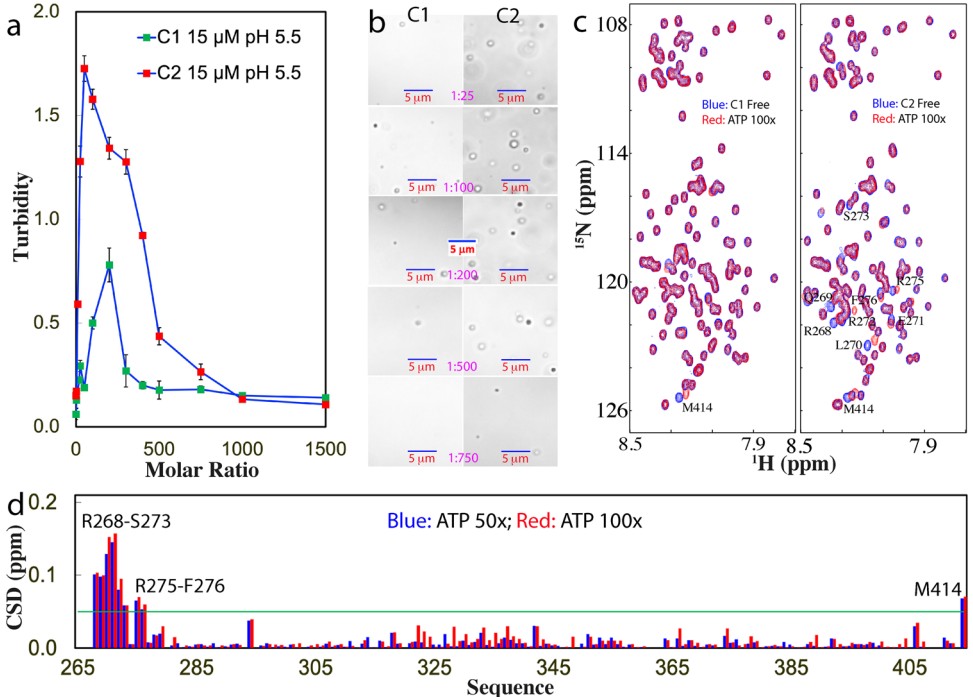

**Fig. 3 DIC and NMR visualization of ATP in modulating LLPS of C1- and C2-PLD. a** Turbidity curves of TDP-43 C1- and C2-PLD at 15 μM in the presence of ATP at different molar ratios in 10 mM sodium phosphate buffer at pH 5.5. **b** DIC images of TDP-43 C1- and C2-PLD at 15 μM in the presence of ATP at different molar ratios in the same buffer at pH 5.5. **c** HSQC spectra of the 15N-labeled TDP-43 C1- and C2-PLD at 15 μM in the same buffer at pH 5.5 in the absence (blue) and in the presence of ATP (red) at a molar ratio of 1:100. **d** Chemical shift difference (CSD) of TDP-43 C2-PLD between in the absence and in the presence ATP at 1:50 (blue) and 1:100 (red). The green line is used to indicate the value: average + SD at 1:100.

rapidly and reach the highest (1.73) at 1:100 which is much higher than that of WT at pH 5.5, but subsequently decreased with further addition of ATP (Fig. 3a). DIC visualization showed that the C2 sample had no droplet or aggregate without ATP but started to form droplets at 1:25. The number of the droplets reached the highest at 1:100 where many droplets were formed. Droplets also became completely dissolved at 1:1000 (Fig. 3b). Noticeably, for C3 with all Arg/Lys mutated to Ala, the transparent sample with aggregates immediately removed by centrifuge would precipitate upon addition of ATP even at 1:10. Moreover, the precipitation still occurred even at a protein concentration of 5 μM upon adding ATP at 1:10. As such, no ATP titrations could be performed on C3 as for C1 and C2 visualized by DIC (Fig. 3b).

We conducted the ATP titrations on three constructs under the same conditions as monitored by NMR HSQC (Fig. 3c). Noticeably for C1 with three N-terminal Arg residues substituted by Ala, even at 1:100 only the HSQC peak of Met414 showed a large shift while the residues clustered over three Arg residues of WT no longer showed large shifts as observed in TDP-43 WT-PLD. At the higher ratio, the HSQC peaks became weaker, and after 1:200 most HSQC peaks became undetectable. By contrast, for C2, at 1:100 in addition to the HSQC peak of Met414, a small set of other HSQC peaks also shifted (Fig. 3c). Detailed analysis revealed that the shifted peaks are all from the residues clustered over the N-terminal three Arg residues whose pattern is very similar to what was observed on WT-PLD titrated by ATP. Further addition of ATP to 1:200 led to the disappearance of its HSQC peaks. Intriguingly for C3, the addition of ATP into the freshly prepared sample even at 1:10 triggered the disappearance of HSQC peaks and the visible precipitation occurred even at a protein concentration of 5 μM. Consequently, no NMR-monitored ATP titrations could be conducted.

The results together clearly suggest that the peak shift of the N-terminal clusters of residues other than three Arg is mostly due to the secondary perturbation induced by the binding of ATP to neighboring three Arg residues because the shift of these residues no longer occurred for C1 with the three Arg residues mutated to Ala. Furthermore, Arg residues appear to play key roles in maintaining the reversibility of LLPS as well as in preventing aggregation of TDP-43 PLD because their mutation all led to the loss of the reversibility of LLPS as well as the proneness to aggregation. Consequently, the results together clearly revealed that ATP modulates LLPS of TDP-43 PLD and its dynamics by specifically targeting Arg residues. It is worthwhile to note that even for C1 and C2 whose droplets could be dissolved as well as no detectable aggregates were formed at high ATP concentrations, at the residue-specific resolution they appear to undergo dynamic association and therefore their HSQC peaks became too broad to be detected.

**LLPS of TDP-43 PLD is controlled by a delicate interaction network**. To understand the contribution of different regions of TDP-43 PLD to LLPS and aggregation upon modulation by ATP, we further generated C4 with the N-terminal residues 265–276 deleted, C5 with Arg293, Arg361, and Lys408 of C4 mutated to Ala as well as C6 with the hydrophobic region 311–343 deleted (Fig. 1a). We assessed the capacity of three variants in phase separation and aggregation at different protein concentrations, buffer pH, and NaCl concentrations. We found that C4 and C5 samples were also prone to aggregation at pH above 6, again highlighting that the aggregation of TDP-43 PLD is not just governed by simple electrostatic interaction. On the other hand, the addition of NaCl even at 50 mM triggered immediate precipitation for C4 and C5, while no precipitation was observed for C6 even with NaCl up to 150 mM. Nevertheless, we have

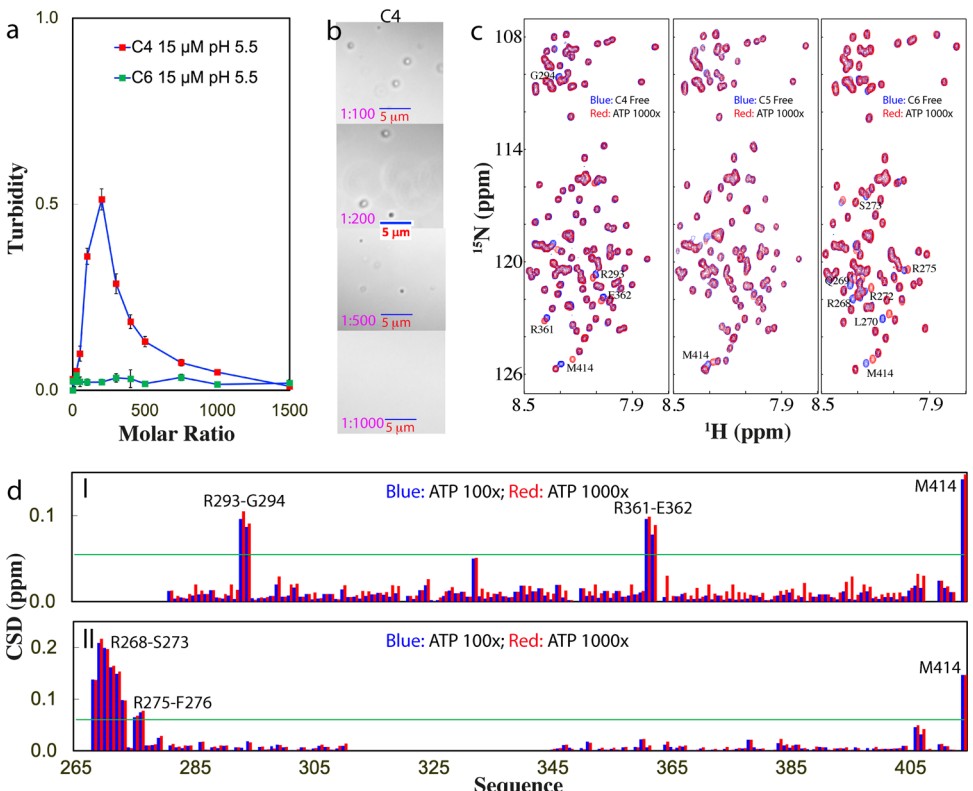

**Fig. 4 DIC and NMR visualization of ATP in modulating LLPS of C4- and C6-PLD. a** Turbidity curves of TDP-43 C4- and C6-PLD at 15 μM in the presence of ATP at different molar ratios in 10 mM sodium phosphate buffer at pH 5.5. **b** DIC images of TDP-43 C4-PLD at 15 μM in the presence of ATP at different molar ratios in the same buffer. **c** HSQC spectra of the $^{15}$N-labeled TDP-43 C4- and C6-PLD at 15 μM, as well as C5-PLD at 10 μM in the absence (blue) and in the presence of ATP at a molar ratio of 1:1000 (red). **d** Chemical shift difference (CSD) of TDP-43 C4- (I) and C6-PLD (II) between the free state and in the presence ATP at 1:100 (blue) and 1:1000 (red). The green line is used to indicate the value: average + SD at 1:1000.

succeeded in acquiring HSQC spectra of the freshly prepared samples of three constructs at 15 μM in 10 mM sodium phosphate buffer (pH 5.5), in which most peaks are superimposable to those of WT (Supplementary Fig. 5), indicating no global changes of their solution conformations.

We then titrated the samples of three constructs at 15 μM with ATP. As shown in Fig. 4a, for C4, without ATP, the turbidity is <0.1 while the addition of ATP induced an increase of turbidity, which reached the highest (0.51) at 1:200. Further addition of ATP led to the decrease of turbidity. By contrast, for C5, the sample without ATP already has a turbidity of 0.19, and the addition of ATP to 1:10 led to visible precipitation. By contrast, for C6, the sample without ATP has a turbidity of 0.023, and the addition of ATP even up to 1:1500 led to no large change of turbidity.

DIC visualization revealed that for C4 (Fig. 4b), without ATP, no droplets were detected while with the addition of ATP at 1:25, some droplets were formed. At 1:200, more droplets could be seen but further addition of ATP led to the reduction of the number of droplets, and at 1:1000, all droplets were dissolved. By contrast, for C5, no droplets were formed without ATP, but the addition of ATP only to 1:10 resulted in visible precipitation. Interestingly, for C6, without ATP no droplets were found and the addition of ATP even upon to 1:1500 failed to induce the formation of any droplets or aggregates.

Interestingly, when we monitored the ATP titrations by NMR HSQC (Fig. 4c), for C4, only several peaks were significantly shifted which were identified to be from Arg293–Gly294 and Arg361–Glu362 (I of Fig. 4d). However, the HSQC peak of Lys408 showed no shift even up to 1:1000, indicating that the binding of ATP to Lys residue is much weaker than that to Arg residue. On the other hand, due to the strong proneness of C5 to

aggregate, we thus reduced the concentration to 10 μM and rapidly added ATP only at two ratios 1:500 and 1:1000. As shown in Fig. 4C, even at 1:1000, no large shifts were detected except for Met414. Remarkably, for C6, although no droplets and aggregates were formed with the addition of ATP up to 1:1500, the residues clustered over the N-terminal three Arg residues were also found to have significant shifts which are largely saturated at 1:100 (Fig. 4c and II of 4d).

The titrations of NaCl into C6 led to no large increase of turbidity (Fig. 5a), as well as no formation of any droplets or aggregates as visualized by DIC. Further HSQC characterization showed that unlike what was observed on other mutants, the addition of NaCl up to 150 mM induced no disappearance of HSQC peaks but only large shifts of a set of HSQC peaks (Fig. 5b). Detailed analysis revealed that the pattern of the shifted residues is also different from that induced by ATP (Fig. 3c). Interestingly, however, the pattern of the shifted residues of C6 induced by NaCl is similar to that of WT induced by NaCl (Fig. 5d). The results clearly indicate that, unlike NaCl which can perturb many different residue types, ATP only specifically binds Arg residues regardless of the sequence contexts of different TDP-43 PLD constructs, again indicating that ATP modulates TDP-43 PLD LLPS by more than simple electrostatic effect. Strikingly, for TDP-43 PLD, whether ATP is able to induce LLPS further depends on the presence of the hydrophobic helical region 311–343, which is previously shown to be the key driver for LLPS of TDP-43 PLD[11,14].

**ATP binds Arg and Lys with different affinity.** To better understand the interaction of ATP with Arg, we generated a

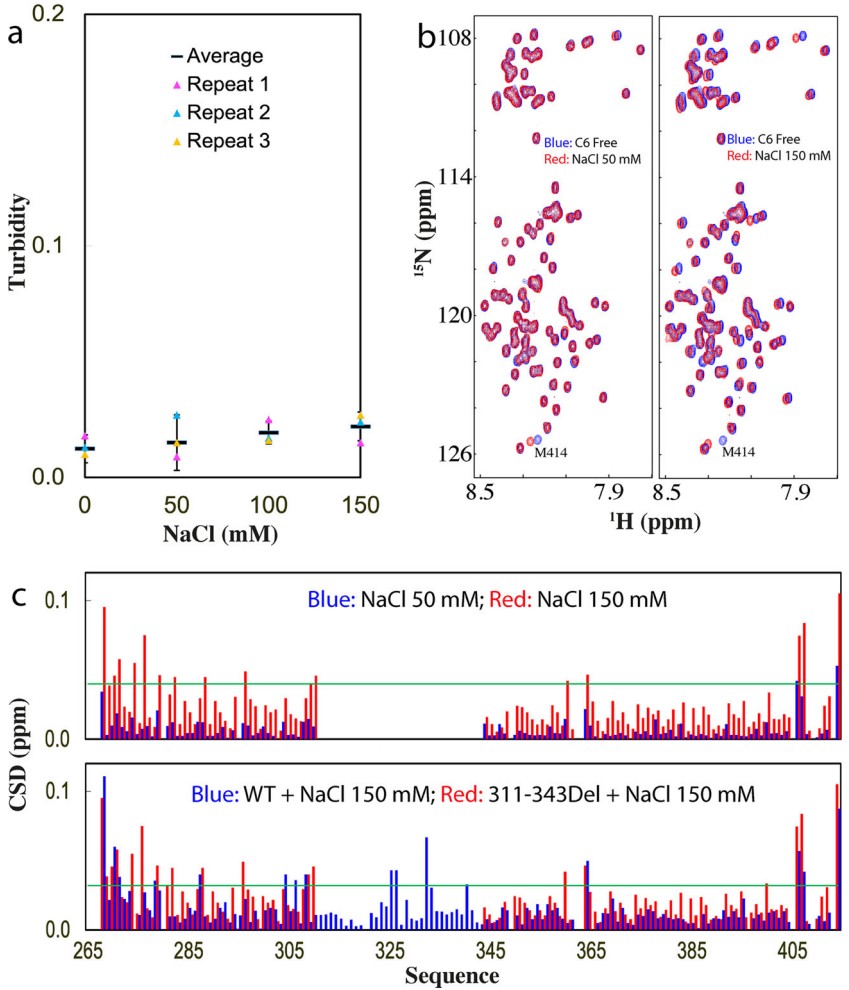

**Fig. 5 NMR characterization of the interaction of C6-PLD with NaCl. a** Turbidity of TDP-43 C6-PLD (three measurements) at 15 μM in 10 mM sodium phosphate buffer (pH 5.5) in the presence of NaCl at different concentrations. **b** HSQC spectra of the $^{15}$N-labeled TDP-43 C6-PLD at 15 μM in the absence (blue) and in the presence of NaCl (red) at 50 and 150 mM, respectively. **c** Chemical shift difference (CSD) of TDP-43 C6-PLD between the free state and in the presence NaCl at 50 mM (blue) and 150 mM (red) (upper) as well as TDP-43 WT-PLD (blue) and C6-PLD (red) in the presence NaCl at 150 mM (lower). The green line is used to indicate the value: average + SD at 150 mM NaCl.

mutant of TDP-43 PLD with all five Arg residues mutated to Lys (C7), which only has the slightly lower pI (9.6) than that (10.78) of WT (Fig. 1a). Very interestingly, C7 could undergo LLPS at 15 μM in the same buffer at pH 5.5 even without ATP, with a turbidity of 0.25 (Fig. 6a) and droplets of ~0.7 μm in diameter (Fig. 6b). The addition of ATP could only induce the slight increase of turbidity which reached the highest of 0.32 at 1:300 (Fig. 6a) as well as the droplet number (Fig. 6b). Further addition of ATP led to a slow decrease of turbidity and the droplet number. However, only at 1:3000, the droplets became completely dissolved with a turbidity of 0.09. On the other hand, at 7.5 μM in the same buffer, C7 showed no phase separation with the turbidity of 0.02 (Fig. 6a), while the addition of ATP only triggered a negligible change of turbidity. Different from what was observed on WT at 7.5 μM which could be induced to form droplets (Fig. 1d and Supplementary Fig. 1c), ATP failed to induce any detectable droplets for C7 at 7.5 μM even up to 1:3000.

C7 has an HSQC spectrum with many peaks superimposable to those of WT (Supplementary Fig. 5), indicating no large change in its overall conformation. We then monitored the ATP titration by HSQC spectroscopy and found that ATP induced the shift of a small set of peaks (Fig. 6c). Intriguingly, similar to what was observed on C2 and C3, upon adding ATP above the ratio of 1:500, most HSQC peaks of C7 become too weak to be detectable,

suggesting that this mutant also lost the reversibility of LLPS. Detailed analysis revealed that C7 has an overall shift pattern very similar to that of WT (Fig. 1f), and the significantly shifted residues include Lys268, Leu270-Lys272, Lys275-Gly277, Ser407-Lys408, and Met414 (Fig. 6d),

Examination of the chemical shift tracings of the significantly shifted peaks of WT and mutants at different ATP ratios disclosed that the shifting of HSQC peaks of WT, C4, and C6 residues were largely saturated at 1:100, while the shifting of all C7 residues except for Met414 remained unsaturated even at 1:500 (Fig. 7a). This clearly indicates that the binding affinity of ATP to Arg is much higher than that to Lys, likely because the purine ring of ATP can establish both π–cation and π–π stacking interactions with the delocalized planar guanidinium cation of Arg but only π–cation interaction with the tetrahedral ammonium cation of Lys. Strikingly, this observation is in general consistent with the previous results that the aromatic rings of aromatic residues[27] and RNA[28] interact with Arg more tightly than with Lys.

## Discussion

Recently, the intrinsically disordered regions (IDRs) have been demonstrated to act as a key driver for LLPS at concentrations much lower than those required for the folded proteins such as

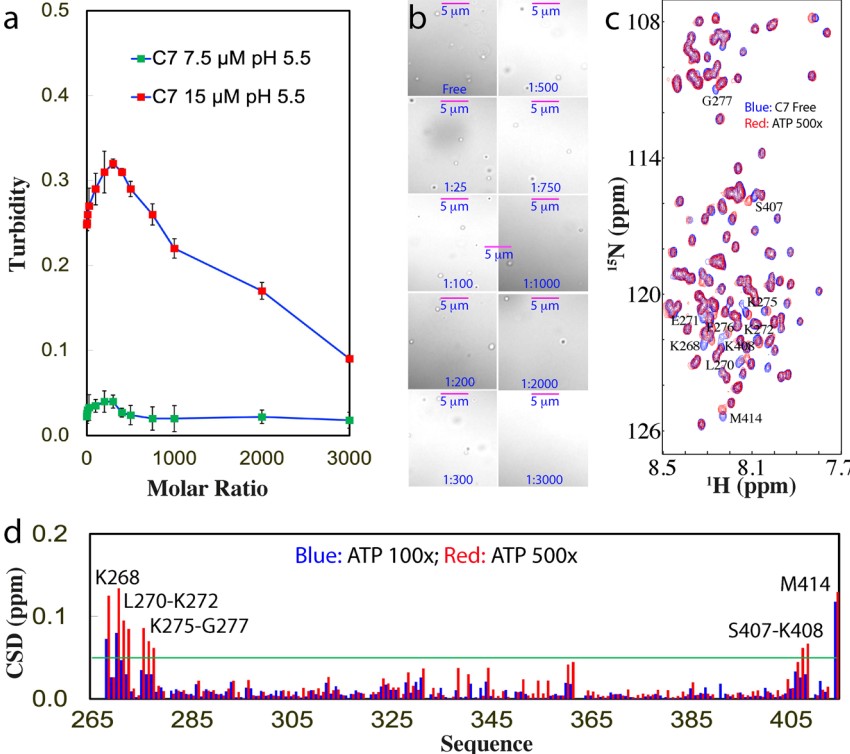

**Fig. 6 DIC and NMR visualization of ATP in modulating LLPS of TDP-43 C7-PLD. a** Turbidity curves of TDP-43 C7-PLD at 15 μM and 7.5 μM in the presence of ATP at different molar ratios in 10 mM sodium phosphate buffer at pH 5.5. **b** DIC images of TDP-43 C7-PLD at 15 μM in the presence of ATP at different molar ratios in the same buffer respective at pH 5.5. **c** $^1$H-$^{15}$N NMR HSQC spectra of the $^{15}$N-labeled TDP-43 C7-PLD at 15 μM in the absence (blue) and in the presence of ATP at a molar ratio of 1:500 (red) in 10 mM sodium phosphate buffer (pH 5.5). **d** Chemical shift difference (CSD) of TDP-43 C7-PLD between the free state and in the presence ATP at 1:100 (blue) and 1:500 (red), respectively. The green line is used to indicate the value (0.05): average + STD at the ATP ratio of 1:500. The residues with CSD values >average + SD are defined as significantly perturbed residues.

lysozyme[29]. Remarkably, LLPS is now emerging as a common principle for organizing intracellular membrane-less organelles underlying cellular physiology and pathology[7–15,18,20,21,30–34]. Initially, the physical framework of the Flory–Huggins theory developed for the phase separation of polymers[35,36] was adapted for describing protein LLPS. Very recently, with respect to the complexity associated with IDRs, a stickers-and-spacers model has been proposed, in which the groups participating in attractive interactions are defined to be "stickers", while the regions between stickers that do not largely contribute to attractive interactions are considered to be "spacers"[32]. Therefore, to identify types, numbers and strengths of stickers represent a key task to elucidate the molecular mechanisms of LLPS of IDRs.

Mysteriously, ATP universally exists at very high concentrations in all living cells. Therefore, the most important finding in the present study is that ATP can biphasically modulate LLPS of TDP-43 PLD: induction at low molar ratios and dissolution at high ratios. Mechanistically, by NMR and site-directed mutagenesis, ATP has been decoded to achieve the modulation by establishing the bivalent interactions with Arg residues, as we previously proposed for the ATP-induced modulation of LLPS of FUS RGG-rich domains[22]. Furthermore, it is very unexpected here to find that only ATP can reversibly modulate LLPS of TDP-43 PLD, while AMP had no capacity and ADP even triggered severe aggregation although ADP also interacts with the same set of TDP-43 PLD residues like ATP. This implies that in addition to the capacity for establishing the bivalent binding, the triphosphate chain of ATP also has some unique capacity, which is essential for reversibly modulating LLPS of TDP-43 PLD. Intriguingly, previous studies including by us showed that ATP is highly dependent on the unique hydration property of the

triphosphate chain in modulating LLPS of proteins including FUS[20–22]. Very recently we have found that although ATP has no detectable binding as well as no effect on the conformation and thermostability of human lens γS-crystallin at a molar ratio up to 1:200 (crystallin:ATP), ATP at 1:1 is unexpectedly sufficient to antagonize the crowding-induced destabilization of γS-crystallin most likely by targeting the protein hydration shell[37,38]. Indeed, by microwave dielectric spectroscopy[24], the triphosphate group has been previously shown to have the "hypermobile water" in addition to the classic "constrained water" (Supplementary Fig. 1a), and one distinguishable difference between ATP, ADP, and AMP is that they have different contents of the "constrained water" and "hypermobile water"[24]. Therefore, in the future, despite extreme challenges, it is of both fundamental and biological interest to explore how the unique hydration property of ATP contributes to the modulation of LLPS and aggregation of TDP-43 PLD, as well as other proteins.

Previously, we were unable to follow the tracings of residue-specific perturbations by NMR for the RGG-rich domain of FUS due to the extensive disappearance of HSQC peaks at most points of ATP titrations. As a result, we could only indirectly derive the speculation that ATP induces and dissolves LLPS of FUS most likely by targeting Arg/Lys residues[22]. Here, we were able to unambiguously conclude that ATP induces and dissolves LLPS of TDP-43 PLD by specifically binding Arg residues as supported by the successful assignment of HSQC spectra of TDP-43 PLD at all ATP concentration points and detailed characterization of seven mutants. In particular, with the all-Lys mutant C7, we obtained the first evidence that ATP specifically binds to Arg with an affinity higher than that to Lys and the high affinity of ATP to Arg is critical for the capacity of ATP in both inducing and

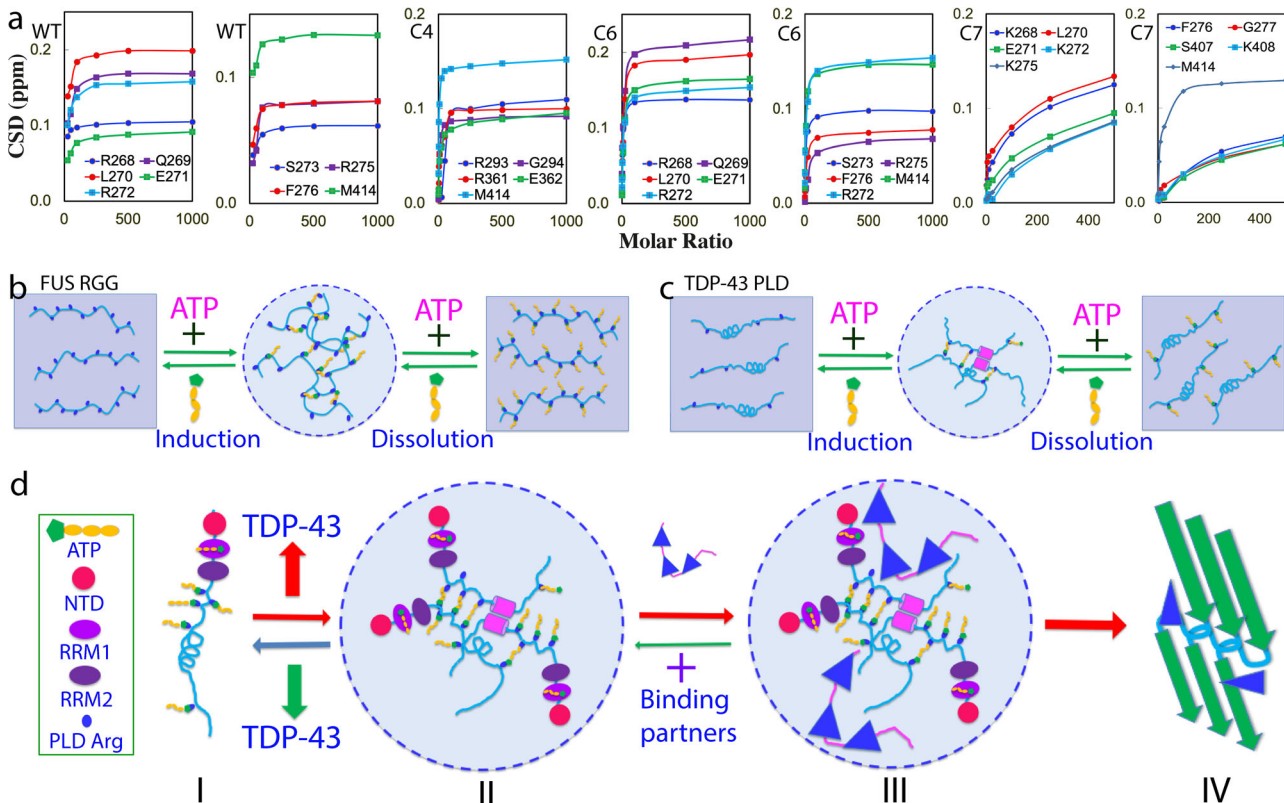

**Fig. 7 Speculative model for ATP to modulate LLPS of cytoplasmic TDP-43. a** Concentration-dependent CSD for the significantly perturbed residues of TDP-43 WT-, C4-, C6-, and C7-PLD. **b** The speculative mechanism for ATP to biphasically modulate LLPS of RGG-rich FUS CTD. **c** The speculative mechanism for ATP to biphasically modulate LLPS of TDP-43 PLD. **d** A speculative model for ATP to modulate LLPS of cytoplasmic TDP-43 and its pathological implications. (I) Under the physiological conditions, LLPS is inhibited due to being bound with ATP of cytoplasmic TDP-43 in neurons, where TDP-43 has concentrations of ~1 μM while ATP has concentrations of ~3 mM. (II) Cytoplasmic TDP-43 reversibly phase separates into dynamic droplets which might be enhanced by the bivalent binding of ATP under pathological/aging conditions with the accumulation of TDP-43, or/and reduction of ATP concentrations. (III) Cytoplasmic TDP-43 droplets become capable of recruiting other proteins. (IV) With a long incubation time, TDP-43 droplets may undergo an irreversible exaggeration from the dynamic droplets into aggregates or/and amyloid fibrils characteristic of ALS.

dissolving LLPS. Therefore, our NMR results on TDP-43 here and FUS before[22] establish that ATP can generally and specifically bind Arg residue within IDRs which do not need to be RGG-/R-rich. Currently, the study only allowed to visualize the effect of ATP on the backbone amides of TDP-43 PLD. In the future, it will be of fundamental interest to detect the direct binding of ATP with Arg side chains, as well as to assess how ATP regulates LLPS and aggregation of the full-length TDP-43.

On the other hand, the exact mechanisms for the ATP binding to modulate LLPS appear to be highly context-dependent. For 156-residue FUS C-terminal RGG-rich domain with 25 Arg residues, which is incapable of LLPS by itself, the bivalent binding of ATP to Arg is sufficient to directly induce LLPS by forming large and dynamic complexes at low ATP ratios followed by dissolution at high ATP ratios due to the exceedingly binding (Fig. 7b)[22]. For 267-residue FUS N-terminal domain (NTD) which can phase separate by itself through the dynamic interaction between Tyr within PLD (1–165) and Arg within RGG1 (166–267)[28,39,40], the binding of ATP to Arg acts to disrupt the existing interaction between Tyr and Arg, thus manifesting as a monotonic dissolution of LLPS[22]. For 150-residue TDP-43 PLD with only 5 Arg residues, the ATP binding is insufficient to solely drive LLPS, and thus needs to coordinate other driving forces[12–14], particularly the dimerization/oligomerization of the unique hydrophobic region[11,14,18] to induce LLPS (Fig. 7c). This is evidenced by the result that the deletion of the hydrophobic region completely abolished the ATP-induced LLPS

although ATP was still able to bind Arg residues in the same manner.

The study also revealed that Arg residues of TDP-43 PLD appear to play a key role in maintaining the reversibility of LLPS as well as in preventing the exaggeration of LLPS into aggregation. Evidently, the mutation of all five Arg to Lys only with a slight change of pI from 10.8 to 9.6 is sufficient to cause the loss of the reversibility of LLPS while the mutation to Ala with a dramatic alternation of pI to 3.6 even leads to severe proneness to aggregation. So what is the underlying mechanism? Very recently it was proposed that the dimeric helix formed over the hydrophobic region[11] might undergo exchanges with the amyloid-like β-rich oligomers particularly at neutral pH, which also contributes to driving LLPS of TDP-43 PLD[18]. Moreover, a slight exaggeration is sufficient to transform the β-rich oligomers into aggregates or/and amyloid fibrils whose structures were determined by Cryo-EM[41]. In this context, the mechanism for Arg residues to maintain reversibility of LLPS and to prevent aggregation of TDP-43 PLD becomes clear. Briefly, as illuminated by the Cryo-EM structure[41], many TDP-43 PLD molecules need to become aligned and stacked up to form oligomers, in which Arg residues of one TDP-43 PLD molecule are in close approximation in space to the corresponding Arg residues of other layers of PLD molecules[41]. Consequently, the net positive charges carried by Arg side chains of different PLD molecules at physiological pH are expected to repulse each other so as to maintain the reversibility of LLPS and to prevent the exaggeration into aggregation/

amyloid fibrils. This mechanism highlights that the reversibility of LLPS and aggregation of TDP-43 PLD are controlled by the conformation-specific electrostatic property, more than simply by the total net charge number of the molecule. As such, unlike FUS in which Arg residues are key "stickers" for driving LLPS by interacting with Tyr, in the context of TDP-43 PLD, Arg residues behave as "inhibitor" for LLPS by providing the conformation-specific repulsive interaction. This thus suggests that the role of Arg in LLPS, whether to serve as a "sticker" or "inhibitor", is highly context-dependent. In the future, it is of general interest to assess whether the context-dependent roles in LLPS also exist for other residues and other proteins.

Previously, nucleic acids including RNA[27,42,43] and ssDNA[22,26] have been shown to induce LLPS at low concentrations but dissolution at high concentrations. A phenomenological model designated as "reentrant phase transition" has been thus proposed that the RNA-induced induction and dissolution of LLPS of Arg/Lys-rich model peptides result from the electrostatic interaction between Arg/Lys and negatively charged phosphate groups of RNA[42]. On the other hand, residue-specific NMR studies indicate that specific π–π/π–cation interactions between Arg/Lys residues and aromatic rings of RNA[28] or ssDNA[22] are also critical for the modulation, which are unexpectedly salt-dependent[44]. Our results here further revealed that in addition to acting as a bivalent binder, ATP can have two different electrostatic effects as a highly charged molecule: alternation of the conformation-specific electrostatic property of IDRs by specific binding to Arg, and non-specific electrostatic screening effect common to all charge molecules including NaCl[45,46]. Previously, the screening effect of salt ions was shown to trigger severe aggregation or even insolubility for unstructured or partially folded proteins with large exposure of hydrophobic patches[45]. Therefore, for all-Lys and all-Ala mutants of TDP-43 PLD to which ATP has weak and even no binding respectively, the screening effect of ATP will become dominant, and consequently ATP acts as NaCl to trigger self-association or even aggregation. This also explains that the perturbation patterns on TDP-43 PLD by ATP and NaCl bear some similarities (Figs. 1f and 2c). However, for WT-PLD to whose Arg residues ATP can achieve the relatively tight binding, ATP additionally alters the conformation-specific electrostatic property of PLD by introducing the highly negatively charged triphosphate. This consequently generates the strong repulsive electrostatic interaction which is long-range and sufficient to suppress the screening effect[46] to maintain the reversibility of LLPS and to prevent exaggeration into aggregation. Therefore, current results bear the critical implication that in addition to the covalent Arg methylation[47–49], ATP might act as a universal and specific regulator for most, if not all, R-containing IDRs by binding to Arg to alter their physicochemical properties, conformations, dynamics, LLPS, assembly, and aggregation. Remarkably, the number of the proteins with R-containing IDRs should be much larger than that of RG/RGG-rich domains which were found in >1700 human proteins[48,49].

Strikingly, the binding of ATP to Arg is largely saturated at the molar ratio of 1:100 for TDP-43 WT-, C4-, and C6-PLD (Fig. 7a). In particular, ATP at 1:1000 is able to dissolve the droplets of TDP-43 PLD at 15 μM while ATP at 1:400 is sufficient to dissolve the droplets of TDP-43 PLD at 7.5 μM. It is established that under normal physiological conditions, the cytoplasm of neurons has ATP concentrations of ~3 mM while TDP-43 is only ~1 μM[18,19,43], thus with a molar ratio of ~1:3000 (TDP-43:ATP). As such, under physiological conditions, TDP-43 is largely bound with ATP over its PLD Arg residues as well as RRM1 domain[50] (I of Fig. 7d). Furthermore, ATP is also a biological hydrotrope capable of dissolving LLPS of RNA-binding proteins at high concentrations[20,21]. Consequently, in the neuronal cytoplasm,

TDP-43 is inhibited for LLPS. Nevertheless, under pathological or/and aging conditions with largely increased TDP-43 or/and reduced ATP concentrations, TDP-43 PLD will drive LLPS which could even be enhanced by the bivalent binding with ATP (II of Fig. 7d). The TDP-43 droplets may become capable of recruiting other proteins into the droplets (III of Fig. 7d) as recently revealed[2]. In particular, as extensively reported[11–14], after a certain period of pathological conditions, dynamic TDP-43 droplets might further exaggerate into irreversible aggregates or/and amyloid fibrils characteristic of ALS (IV of Fig. 7d), thus ultimately leading to cell death[2].

In summary, by the specific and relatively tight binding to Arg residues, ATP achieves the biphasic modulation of LLPS of TDP-43 PLD. As ATP, the universal energy currency exists in all cells with concentrations of 2–12 mM, our study logically establishes that in addition to the covalent Arg methylation, ATP might act as a general but specific regulator for most, if not all, R-containing IDRs by binding to Arg residues to alter their physicochemical properties, conformations, dynamics, LLPS, assembly, and aggregation.

## Methods

**Preparation of recombinant WT and mutated TDP-43 PLD proteins**. In this study, we utilized our previously-cloned DNA construct encoding TDP-43 265-414 (PLD) and transferred it into a modified vector without any tag[16,22,26]. Six mutants of TDP-43 PLD were further generated by the use of QuikChange Site-Directed Mutagenesis Kit (Stratagene, La Jolla, CA, USA)[16].

All eight recombinant TDP-43 PLD proteins were highly expressed in *E. coli* BL21 cells but all were found in inclusions. Consequently, their purification followed the previously established protocols by others and our labs. Briefly, the recombinant proteins were solubilized with the buffer with 8 M urea, and the reverse phase (RP)-HPLC purification was used to obtain highly pure proteins with the impurities including liquid, ions, and nucleic acids removed[12,16,22,26]. Isotope-labeled proteins for NMR studies were prepared by the same procedures except that the bacteria were grown in M9 medium with the addition of $(^{15}NH_4)_2SO_4$ for $^{15}N$-labeling[16,22,26]. The protein concentration was determined by the UV spectroscopic method in the presence of 8 M urea, under which the extinct coefficient at 280 nm of a protein can be calculated by adding up the contribution of Trp, Tyr, and Cys residues[16,20,22,26,51].

ATP, ADP, and AMP were purchased from Sigma-Aldrich with the same catalog numbers as previously reported[19–21,50]. The protein and ATP samples were all prepared in 10 mM sodium phosphate buffer, and $MgCl_2$ at the equal molar concentration to ATP was added for stabilization by forming the ATP–Mg complex[19–21,50]. The final solution pH values were checked by pH meter and the small variations were adjusted with aliquots of very diluted NaOH or HCl.

**Differential interference contrast (DIC) microscopy and turbidity measurement**. The formation of liquid droplets and aggregates was imaged at 25 °C on 50 μl of different TDP-43 PLD samples at various protein concentrations in 10 mM sodium phosphate or acetate buffers at different pH values and in the absence and in the presence of ATP at molar ratios of 1:0, 1:2.5, 1:10, 1:25, 1:50, 1:100, 1:200, 1:300, 1:400, 1:500, 1:750, 1:1000, and 1:1500 by differential interference contrast (DIC) microscopy (OLYMPUS IX73 Inverted Microscope System with OLYMPUS DP74 Color Camera)[20,24]. NaCl was titrated into different construct samples to reach 50, 100, and 150 mM, respectively. The turbidity measurement and DIC imaging were performed after 15 min of the sample preparation. The turbidity was measured three times at a wavelength of 600 nm and reported as average + SD.

**NMR characterizations**. All NMR experiments were acquired at 25 °C on an 800 MHz Bruker Avance spectrometer equipped with pulse field gradient units and a shielded cryoprobe[16,22,23,50]. To have the enhancing effect of the cryoprobe for NMR signal sensitivity, which is essential for NMR HSQC titration experiments at such a low protein concentration (15 μM), NMR samples had to be prepared in 10 mM sodium phosphate buffer, while pH value was optimized to 5.5 as many HSQC peaks of TDP-43 PLD disappeared at higher pH values due to the enhanced exchange with bulk solvent and/or dynamic association.

For NMR titration studies of the interactions between TDP-43 WT/mutated PLD proteins and ATP, ADP, and AMP, one-dimensional proton and two dimensional $^{1}H$-$^{15}N$ NMR HSQC spectra were collected on $^{15}N$-labeled samples at a protein concentration of 15 μM in 10 mM sodium phosphate buffer (pH 5.5) at 25 °C in the absence and in the presence of ATP, ADP, and AMP at molar ratios of 1:0, 1:2.5, 1:10, 1:25, 1:50, 1:100, 1:200, 1:300, 1:400, 1:500, 1:750, and 1:1000.

NMR data were processed by NMRPipe[52] and analyzed by NMRView[53]. To calculate chemical shift difference (CSD) induced by interacting with ATP and NaCl, HSQC spectra were superimposed and subsequently the shifted peaks were

identified, which were further assigned to the corresponding residues of TDP-43 PLD with the NMR resonance assignments previously achieved by us and other groups[11,12,16]. The degree of the perturbation was reported by an integrated index calculated by the following formula:[22,23,50,54].

$$CSD(ppm) = ((\Delta^1 H)^2 + (\Delta^{15} N)^2 / 4)^{1/2} \quad (1)$$

The residues with CSD values > average + STD are defined as significantly perturbed residues.

**Statistics and reproducibility**. For NMR and DIC experiments, the exploratory experiments of TDP-43 PLD and its seven mutants titrated with ATP, ADP, AMP, and NaCl at different concentrations were first conducted to identify the optimized concentration ranges. Subsequently, the final DIC and HSQC titrations were performed once with the optimized points of ATP, ADP, AMP, and NaCl concentrations.

**Reporting summary**. Further information on research design is available in the Nature Research Reporting Summary linked to this article.

## Data availability
The data supporting the findings of this study are available within the paper and Supplementary Data 1. All other data are available from the corresponding author upon reasonable request.

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

## Acknowledgements

This study is supported by the Ministry of Education of Singapore (MOE) Tier 1 Grant R-154-000-B92-114 to Jianxing Song. The funders had no role in study design, data collection and analysis, decision to publish, or preparation of the manuscript.

## Author contributions

J.S. D.M., L.L., and J.K. conceived and designed the experiments. D.M., L.L., J.K., and J.S. performed the research, analyzed the data, J.S. wrote and revised the manuscript.

## Competing interests

The authors declare no competing interests.
