## [Transparent Peer Review File · Communications Biology]

Reviewer #1 (Remarks to the Author):

Summary

This paper by Mei et al. provides a molecular understanding of how ATP interacts with the PLD of TDP-43, a RNA-binding protein that is implicated in ALS and other neurological disorders. This work illustrates that ATP acts to both promote and inhibit phase separation of TDP-43 PLD as a function of increasing ATP concentration -- this is very reminiscent of re-entrant phase separation behavior that has been recently seen for RNA-protein coacervate systems (recent work by Priya Banerjee and others, and also see <https://www.biorxiv.org/content/10.1101/2020.05.04.076299v2>). Using NMR spectroscopy and various mutation and deletion constructs, the authors show that Arg residues are implicated in binding ATP. Deletion of Arg residues or Arg mutations to Ala modulate TDP-43 PLD LLPS propensity with ATP. These effects are particular to ATP as ADP and AMP do not elicit the same effect. This work is interesting and important to elucidating how ATP affects LLPS of biomolecular systems with speculation on ATP effects inside a cell. However there are several concerns and experiments that need to be addressed before recommendation for publication. Several statements (including one in the title) overextend and are not supported by the data. These need to be dialed back, as noted below. Additionally, the discussion of inhibitory interactions needs to be revamped (see below).

Major:

- 1) The Arg-to-Ala mutations yield interesting turbidity assay and NMR results, but side effects include changes in protein pI and aggregation propensity (the latter comes up with quite a few variants presented here). Can the authors redo experiments using Arg-to-Lys mutations that would lessen the effect on overall pI and perhaps protein aggregation as well?
- 2) All of the experiments here used a single PLD protein concentration to monitor effects of ATP on LLPS. A second protein concentration should be used (30 μ M or 7.5 μ M, for example), to see how the ATP trends vary, particularly as the PLD self-associates. A possible alternative interpretation is that ATP is modulating PLD's ability to self associate, and not just binding to Arg residues? This is based on comparing Figure 1E to Figure S3C. The CSD pattern in Figure S3C mimics part of Figure 1E, and it may be that the added NaCl may accentuate PLD self-association via enhanced hydrophobic interactions, for example. Use of Arg-to-Lys mutants may also lessen propensity of the protein to aggregate, thus enabling these experiments.

Textual "big" corrections:

- 1) Remove ALS-initiating from the title. This work is entirely *in vitro*, and the connection to ALS is not warranted by the data. Additionally, ALS-linked mutations are not discussed in this manuscript.
- 2) Reword the discussion about "inhibitors" in the context of sticker and spacer effects. It is not clear what is meant by how arginine residues are inhibitors of LLPS: "Arg residues, the key sticker for LLPS of FUS, have been identified here to act as the inhibitor for LLPS of TDP-43". It is increasingly appreciated that amino acids elicit different effects on LLPS depending on sequence context, sequence composition, etc. Instead of calling Arg residues inhibitors, it is more appropriate to call them modulators of LLPS in the context of TDP-43, particularly in the absence of information for how the 5 Arg residues interact with other parts of TDP-43 on an intramolecular or intermolecular level.
- 3) It is not clear why "biphasic modulation" is used here. Why not call this re-entrant phase separation? (see: <https://www.ncbi.nlm.nih.gov/pmc/articles/PMC5976450/>)
- 4) Discuss mechanism of why ATP causes re-entrant phase separation for the general reader.

Corrections:

- 1) Figure 1B - create a new figure where turbidity data for all constructs are on a single plot to improve clarity.
- 2) Need to show microscopic images of the aggregates formed by C2, C3, C4 etc.
- 3) Pg. 9 "proneness of C3 with pI of \sim 3.6" should be referring to C4?

- 4) Figure S1B could use a legend. Also, description of colors is inconsistent in referring to line or marker color.
- 5) Should add No ATP pictures to Fig 1C to provide evidence for lack of LLPS in addition to turbidity data.
- 6) They cite their own 2019 work saying "ATP also mediates LLPS of intrinsically-disordered RGG-rich domain" on page 4 line 4. "Mediates" implies that ATP brings LLPS about and is required. However, this is only true of the CTD of FUS. There are 2 RGG-rich domains; RGG1 (NTD side) separates without ATP, RGG2 (when attached to CTD) does not. These differences suggest that either homotypic and heterotypic interactions drive LLPS of FUS, and that should be mentioned here.

There are numerous editorial comments as there were many sections where language was unclear, and some are listed here (however this list is not comprehensive):

Pg 2 lines 6 and 8 - in the numbered results "its" should be "TDP-43 PLD" as TDP-43 is not already a subject in those statements and the sentences were a bit confusing to understand.

Pg 3 First paragraph, last sentence needs citation

Pg 3 Second paragraph, last sentence needs citation and "undergoes the toxic LLPS"- does not need "the".

Same sentence as above - Not clear what "toxic LLPS" means, and its cause-result order with ALS is not clear.

Pg 4 Citation 19 does not mention TDP-43, need new citation

Pg 4 First paragraph second last sentence [ATP] in cells should have citation.

Pg 4 Second paragraph line 9 "conjecture" should be "conjunction" if that's what they mean

Pg 6 First paragraph, last sentence, "exaggerate into aggregation" needs rephrasing

Pg 6 Last sentence needs data or citation

Pg 7 First paragraph last sentence "became disappeared"- awkward phrasing. Delete became.

Pg 16 Last line "conjecture"- does not fit context, maybe "conjunction" instead

Multiple pages - "even not RGG-/R-rich" - awkward phrasing, perhaps "that are not RGG-/R-rich" if that is the correct meaning.

The article by Mei et al. reports that ATP biphasically modulates the liquid-liquid phase separation (LLPS) of the disordered prion-like domain (PLD) of TAR DNA-binding protein 43 (TDP-43). The authors used DIC microscopy and NMR spectroscopy to demonstrate that ATP induces the LLPS of TDP-43 PLD at low molar ratios and consequently dissolves its LLPS at high molar ratios. For their studies, the authors used seven constructs of TDP-43 PLD, including WT and six mutants. Based on their data, the authors claim that ATP achieves the biphasic modulation by specifically binding to the Arg residues. Additionally, the authors report that both the Arg residues and the hydrophobic region play critical roles in controlling the LLPS and aggregation of TDP-43 PLD.

The manuscript is logically-structured; the data are clearly presented. The data support their main conclusion. However, there are many technical flaws that need to be corrected before the manuscript could be considered for publication. The authors will need to address several major and minor concerns, as given below. With attention to these issues, the manuscript may be suitable for further consideration by the Communications Biology Journal; the topic of the studies will certainly be of interest to the Communications Biology journal's readership.

MAJOR CONCERNS:

-Throughout the manuscript, the authors claim several observations for which they do not provide any experimental data. The authors need to present these data, at least in the supplementary information. Without the experimental data, it's hard to believe their observations/results. Also, a prevalent weakness of the manuscript is a lack of rigor in assessing the phase separation behavior of the multiple TDP-43 PLD constructs that are reported. This is done largely on the basis of non-quantitative analyses of DIC microscopy images. A rigorous approach would be to determine C_{sat} values for the different constructs by varying protein concentration and establishing quantitative image analysis criteria that define when phase separation/aggregation occurs. Additionally, the authors should provide phase diagrams either by turbidity assay or DIC microscopy images for TDP-43 PLD LLPS in the absence and presence of ATP for the readers to appreciate the change in the C_{sat} values.

-Following is the list of the author's observations, for which they do not show any experimental data in the manuscript.

- 1) In the first section of the results (first paragraph, Page 6), "concentrations $> 50 \mu\text{M}$, TDP-43 PLD phase separates without ATP but undergo aggregation after 1 h".
- 2) In the first section of the results (second paragraph, Page 6), "DIC images showed that without ATP, no droplets or aggregates were formed".
- 3) In the first section of the results (second paragraph, Page 6), "the DIC image at 1:1000 showed dissolution of droplets".

- 4) In the second section of the results (third paragraph, Page 8), “DIC images indicated that no droplets or aggregates were detectable at all points of AMP concentrations”.
- 5) In the second section of the results (fifth paragraph, Page 8), “DIC images indicated that no droplets or aggregates were formed in the presence of NaCl even up to 150 mM”.
- 6) In the third section of the results (second paragraph, Page 9), “All of the mutants (C2, C3, and C4) become severely prone to aggregation at pH 6.0”.
- 7) In the third section of the results (second paragraph, Page 9), “At pH 5.5, the C2 and C3 samples showed no significant aggregation at 15 μ M in 10 mM phosphate buffer.”
- 8) In the third section of the results (third paragraph, Page 10), “DIC images showed that without ATP, the C2 sample had no droplets or aggregates. Also, droplets got completely dissolved at 1:1000 molar ratio”.
- 9) In the third section of the results (fourth paragraph, Page 10), “DIC images indicated that the C3 sample had no droplets or aggregates without ATP. Also, for the C4 sample, the addition of ATP at 1:10 induced precipitation.
- 10) In the third section of the results (fifth paragraph, Page 10), “Conducted the ATP titrations on three constructs, i.e., C2, C3, and C4. However, the data for the C4 construct is not shown in the manuscript.
- 11) In the fourth section of the results (third paragraph, Page 12), “DIC images revealed that for C5, without ATP, no droplets were detected while with the addition of ATP at 1:25, some droplets were formed”.
- 12) In the fourth section of the results. The authors do not show the DIC images for both the C6 and C7 mutants.

MINOR CONCERNS:

- 1) Figure 1- Panel A-The authors should replace 1, 2, 3, 4, 5, 6, 7 with WT, C2, C3, and so on because the authors have used the latter notations for referring to different constructs in the manuscript. Also, please mention in the figure caption that the numbers written in blue are the pI values. In Panel B, the authors should mention the wavelength at which the turbidity values were measured (on the y-axis). Also, the colors used are confusing here. I would suggest using different symbol colors and the same line color. This same comment is for Figure S1B as well.
- 2) In the introduction (first paragraph, Page 3), “...despite being intrinsically-disordered, TDP-43 PLD host almost all ALS-causing mutations...” And, (second paragraph, Page 3), “...cytoplasmic TDP-43 droplets act to recruit importin- α and Nup62....” The authors need to provide the references here.

- 3) In the introduction (third paragraph, Page 4), “We further found that by specific binding, ATP also mediates LLPS...” The authors should rephrase this sentence as this observation is from their previous findings.
- 4) In the introduction (fourth paragraph, Page 4), Suggest rephrasing: “For the first time, we found that ATP is indeed capable...”
- 5) In the first section of the results (first paragraph, Page 6), there is a typo (25 °C). The authors should also specify whether they have used NaCl in their studies (in Figure 1, Figure 2, etc.). If yes, then the authors should mention the concentration, either in the figure caption or in the Methods. And, (second paragraph, Page 6). suggest rephrasing: “... 1.62 and 0.94 respectively at pH 7.0 and 5.5”.
- 6) In the second section of the results (second paragraph, Page 7), “As it is well-documented that the C-terminal.....” The authors need to provide the reference here.
- 7) In the second section of the results (fourth paragraph, Page 8), the authors state that the DIC images of TDP-43 PLD in the presence of ADP at 1:200 molar ratio induced the formation of detectable aggregates. However, in the micrographs, aggregates are not visible at this ratio. The authors should comment on this.
- 8) The authors report rapid aggregation of a few of the mutant constructs, specifically the ones lacking all of the Arg residues. Do the authors know whether these aggregates are amorphous or amyloid-like? The authors should comment on this in the manuscript.
- 9) The scale bars should be provided for all of the DIC images (for an individual image), and the details can be specified in the figure caption in case the space is less.
- 10) In the figure caption 2C, please mention that the data in the presence ATP is shown in red.
- 11) In the figure caption 3D, either write upper & lower or I & II to refer to the TDP-43 C5-and C7-PLD.
- 12) In the figure caption S1B, please mention the molar ratio of TDP-43 PLD and ATP/ADP/AMP used for these studies.
- 13) In the figure caption S3B, please specify which data is shown in blue.
- 14) Many of the references are erroneously formatted. The authors may want to export article citations directly from PubMed using EndNote or some other reference management software to standardize their format.

Reviewer #3 (Remarks to the Author):

In this paper named “ATP biphasically modulates ALS-initiating LLPS by specifically binding to Arg, an inhibitory network component of TDP-43 PLD” Mei et al. showed how ATP modulates the liquid-liquid phase separation (LLPS) of TDB-43 PLD using DIC and NMR. In this study, the authors investigated

the effect of ATP, ADP, and AMP on the LLPS of WT TDP-43 PLD and its six variants. At a low concentration of ATP, TDB-43 PLD forms LLPS and subsequent addition of ATP to this solution dissolves that LLPS. TDP-43 PLD only contains five Arg residues. Authors showed that ATP specifically binds to these Arg residues at a particular molar ratio to modulate its LLPS. Later on, its aggregation is regulated by a delicate network composed of both attractive and inhibitory interactions between those Arg residues and ATP. Definitely, along with this bivalent binding, ATP has some unique hydration property to reverse the LLPS of TDP-43 as it showed contrasting behavior to AMP. Although the application of ATP in modulating LLPS is not new (Hayes et al life 2018), the atomic level information from the NMR study on TDP-43 PLD will be useful to understand the molecular mechanism of the LLPS and the aggregation of this protein. This kind of study will give new ideas to get rid of the deleterious effect of aggregation by treating only the LLPS. The paper is well written but there are few suggestions to the authors to improve the manuscript:

1) On page 6, a reference is required ".....On the other hand, at concentrations > 50 μ M, it could phase separate without ATP but would exaggerate into aggregation after 1 hr."

2) On page 7, the authors mentioned about "that the number of the droplets was higher than that at pH 5.5 (Fig. 1C), thus resulting in the higher turbidity (Fig. 1B)." I am wondering about the pI of the wild-type TDP-43 PLD protein and if higher pH induces more aggregation, then is there any correlation between the number of droplets and aggregation propensity? What would be the author's comment about it?

3) In figure 1C, the Image scale is missing. I am assuming that the scales are the same. Then, is the size of the droplets at pH5.5 different from pH 7.0? Did authors see any different droplet fusion at these two pHs?

4) In figure 1D (1H- 15N NMR HSQC spectra), few red peaks looked broader or may be clustered. What do authors think whether it may be for protein aggregation or due to rapid amide proton exchange with solvent or fast protein dynamics at 1:1000 molar ratio?

5) On page 9, probably, the authors wanted to mean C4 instead of C3 in this sentence ".....it is unexpected for the proneness of C3 with pI of \sim 3.6".

6) This is a very interesting observation for C4 as they mentioned "Noticeably, even for the transparent C4 sample immediately with aggregates removed by centrifuge, the addition of ATP at 1:10 induced precipitation as detected by DIC. Moreover, the precipitation still occurred even at a protein concentration of 5 μ M upon adding ATP at 1:10." In the C4 construct, the authors deleted all the key Arg residues. What will be the molecular mechanism of this observation or which interactions are responsible for the formation of this turbidity in the context of these Arg residues?

7) The authors did not mention how long did they wait before taking their reading in DIC and NMR? I am wondering about the lifetime of these droplets.

8) In figure 2B, the authors marked C2 and C3 columns in blue as "Nm-Cw and Nw-Cm". The authors did not mention about them.

9) Authors used 5uM for the DIC experiment and for most of the NMR experiment, they used 15uM. Is there any reason for using two different concentrations for these two experiments?

10) I am wondering whether authors should give the number (III) image for their speculative model as they only established (I),(II), and (IV) by their experiments. I understood that they tried to cover the bigger picture but, they did not do any experiment with Nup62 and Importin- α .

11) It will be helpful if authors will provide a reference for this ".....it rationalizes the observed susceptibility of the intrinsically-disordered domains to minor perturbations such as the Met337Val

mutation of TDP-43 PLD, which is sufficient to significantly alter the properties of its LLPS and aggregation.”

12) M414 showed significant chemical Shift difference (CDS) (ppm) for almost all the construct that the authors used. What will be the author’s interpretation of the role of M414 in the formation of LLPS? Has M414 any interaction with ATP or is M414 a part of the delicate network in the cytoplasmic TBP-43 sequence?

13) Following question 6 again, in the manuscript, authors showed that one variant without Arg residues showed a high propensity to aggregates, whereas, other five variants showed the same trend of turbidity at low concentration of ATP and then the disappearance of this LLPS at high concentration of ATP. Thus, the manuscript has a very interesting observation of the role these five Arg residues to form LLPS and then aggregates. Thus, for the LLPS formation, these Arg residues are required but what is about aggregation (like their C4 construct)? In the discussion, an explanation of the connection between LLPS and aggregation in the context of these variants will be more interesting to the readers.

14) I am not aware of the journal reference policy but the authors gave 9 self-citations out of 43 references, which is nearly 21%.

Point-to-point Response

Reviewer 1.

Major:

1) The Arg-to-Ala mutations yield interesting turbidity assay and NMR results, but side effects include changes in protein pI and aggregation propensity (the latter comes up with quite a few variants presented here). Can the authors redo experiments using Arg-to-Lys mutations that would lessen the effect on overall pI and perhaps protein aggregation as well?

Response: Thanks so much for this critical suggestion.

We have now constructed the all-Lys (C7) mutant and conducted the parallel experiments on it. Indeed, the new results on this mutant not only addressed the comments by the reviewers, but together with the results in a new PNAS paper (Ref.18) also allowed us to propose a speculative mechanism to rationalize the role of TDP-43 PLD Arg residues in LLPS and aggregation. More generally, these results also offer the first insight into the difference of the binding affinity of ATP to Arg and Lys within IDRs, thus bearing a critical implication in understanding the molecular interactions for ATP functions at mM.

In the revised manuscript, the obtained results were summarized in a new figure (Figure 6) and a new section was developed to describe the results. Moreover, extensive revisions have been conducted over the whole main text to discuss these results.

2) All of the experiments here used a single PLD protein concentration to monitor effects of ATP on LLPS. A second protein concentration should be used (30 μ M or 7.5 μ M, for example), to see how the ATP trends vary, particularly as the PLD self-associates. A possible alternative interpretation is that ATP is modulating PLD's ability to self associate, and not just binding to Arg residues? This is based on comparing Figure 1E to Figure S3C. The CSD pattern in Figure S3C mimics part of Figure 1E, and it may be that the added NaCl may accentuate PLD self-association via enhanced hydrophobic interactions, for example. Use of Arg-to-Lys mutants may also lessen propensity of the protein to aggregate, thus enabling these experiments.

Response: Thanks so much for these comments.

Previously after the preliminary optimization of buffer conditions and protein concentrations, for each construct we only performed one complete set of experiments at one protein concentration because even to perform one set of experiments with many points of ATP concentrations required a large amount of protein samples. On the other hand, some mutants had very low expression level in *E. coli* cells. Nevertheless, to address the comment, we have now increased another protein concentration (7.5 μ M). Interestingly, for the WT-PLD at 7.5 μ M, ATP can still induce LLPS, while for all-K mutant, ATP is no longer able to induce LLPS. The detailed description and discussion were incorporated into the revised manuscript.

With the new results on all-Lys mutant, we were now able to differentiate the electrostatic effects of ATP on TDP-43 PLD into two different types: 1) the conformation-dependent alternation of the electrostatic property of TDP-43 PLD caused by the specific ATP binding to Arg; 2) the electrostatic screening effect common to all charged molecules including ATP and NaCl. As such, the CSD

patterns induced by ATP and NaCl bear some similarity, as kindly pointed out by the reviewer.

To adequately address this point, we have moved two supplementary figures on NaCl to Figure 2 and Figure 5 respectively. Furthermore, extensive revisions have been made in Results and Discussion.

Textual “big” corrections:

1) Remove ALS-initiating from the title. This work is entirely in vitro, and the connection to ALS is not warranted by the data. Additionally, ALS-linked mutations are not discussed in this manuscript.

Response: Thanks so much for this suggestion and we have removed them.

2) Reword the discussion about “inhibitors” in the context of sticker and spacer effects. It is not clear what is meant by how arginine residues are inhibitors of LLPS: “Arg residues, the key sticker for LLPS of FUS, have been identified here to act as the inhibitor for LLPS of TDP-43”. It is increasingly appreciated that amino acids elicit different effects on LLPS depending on sequence context, sequence composition, etc. Instead of calling Arg residues inhibitors, it is more appropriate to call them modulators of LLPS in the context of TDP-43, particularly in the absence of information for how the 5 Arg residues interact with other parts of TDP-43 on an intramolecular or intermolecular level.

Response: Thanks so much for this suggestion. With the new results on all-K mutant and a latest PNAS paper from McKnight’s group (Ref. 18), we were able to propose a speculative mechanism for Arg to maintain the reversibility of LLPS and to prevent the exaggeration into aggregation. Furthermore, we extensively revised the discussion as the reviewer kindly suggested.

3) It is not clear why “biphasic modulation” is used here. Why not call this re-entrant phase separation? (see:

<https://www.ncbi.nlm.nih.gov/pmc/articles/PMC5976450/>)

Response: Thanks so much for this suggestion.

We have read these literatures very carefully and also added them into our reference list. As evidenced by these literatures, “re-entrant phase separation” is a phenomenological model emphasizing the observation that some proteins can have more than one phase separation states upon altering salt or RNA concentrations, not necessarily referring to induction followed by dissolution of LLPS by one factor as we observed on ATP. Furthermore, the original proposal on LLPS of RNA-model-protein system is that “re-entrant phase separation” resulted from non-specific electrostatic effect of the phosphate group of RNA. On the other hand, however, based on NMR studies on RNA-induced LLPS of tau by Zweckstetter’s group (Ref.29) as well as on ssDNA/RNA-induced LLPS of FUS by us, the specific bivalent interactions of ATP/RNA/DNA between aromatic rings of ATP/RNA/DNA with Arg/Lys as well as between phosphate groups with Arg/Lys appear to play key roles.

As such, here we used “biphasic” to highlight the dual effects of ATP more than the multiple possibility of phase separation states of proteins. This appears to be a general capacity of ATP as we now found that ATP not only has such a biphasic effect on LLPS of FUS and TDP-43 PLD, but also on LLPS of the SARS-CoV-2 N protein. So at the current stage it would be informative and avoiding the

direct conflict between the residue-specific NMR results and phenomenological model.

4) Discuss mechanism of why ATP causes re-entrant phase separation for the general reader.

Response: Thanks so much and we have enhanced the discussion on this point.

Corrections:

1) Figure 1B - create a new figure where turbidity data for all constructs are on a single plot to improve clarity.

Response: Thanks for this suggestion. I have made the plot with all curves in one plot but they are too crowded to be seen clearly. To solve this problem, I have placed the curve plots of all constructs in Fig. S1C with the same scales.

2) Need to show microscopic images of the aggregates formed by C2, C3, C4 etc.

Response: Thanks so much for this comment.

In fact, my group has been focused on studying aggregation and fibrillation of TDP-43 PLD since 2010 and published the first systematic study on aggregation and amyloid fibrillation of the full-length TDP-43 PLD and its ALS-causing mutants (Lim et al., Song [2016] *PLoS Biology*. Ref. 16).

In all previous studies, we found that once the samples become quickly aggregated, they looked more or less similar as imaged by DIC and EM which were almost indistinguishable for different mutants. Only under the optimized buffer conditions and protein concentrations, the different properties of the formed amyloid fibrils could be visualized for WT and mutants. As now the aggregation of TDP-43 PLD has been exhaustively studied by many groups including us, as well as the aggregation is not the focus of the current study, here I presented some typical images of aggregates in Fig. S1B.

3) Pg. 9 “proneness of C3 with pl of ~3.6” should be referring to C4?

Response: Thanks so much for pointing out this mistake.

As initially we named WT to be C1 but later changed, there existed several mistakes with the numbering. We have now corrected them all.

4) Figure S1B could use a legend. Also, description of colors is inconsistent in referring to line or marker color.

Response: Thank you so much for this suggestion and we have corrected it.

5) Should add No ATP pictures to Fig 1C to provide evidence for lack of LLPS in addition to turbidity data.

Response: Thanks so much and we have added them.

6) They cite their own 2019 work saying “ATP also mediates LLPS of intrinsically-disordered RGG-rich domain” on page 4 line 4. “Mediates” implies that ATP brings LLPS about and is required. However, this is only true of the CTD of FUS. There are 2 RGG-rich domains; RGG1 (NTD side) separates without ATP, RGG2 (when attached to CTD) does not. These differences

suggest that either homotypic and heterotypic interactions drive LLPS of FUS, and that should be mentioned here.

Response: Thanks so much for this comment and we have added a short discussion on ATP effect on FUS NTD in the revised manuscript.

There are numerous editorial comments as there were many sections where language was unclear, and some are listed here (however this list is not comprehensive):

Pg 2 lines 6 and 8 - in the numbered results “its” should be “TDP-43 PLD” as TDP-43 is not already a subject in those statements and the sentences were a bit confusing to understand.

Pg 3 First paragraph, last sentence needs citation

Pg 3 Second paragraph, last sentence needs citation and “undergoes the toxic LLPS” - does not need “the”. Same sentence as above - Not clear what “toxic LLPS” means, and its cause-result order with ALS is not clear.

Pg 4 Citation 19 does not mention TDP-43, need new citation

Pg 4 First paragraph second last sentence [ATP] in cells should have citation.

Pg 4 Second paragraph line 9 “conjecture” should be “conjunction” if that’s what they mean

Pg 6 First paragraph, last sentence, “exaggerate into aggregation” needs rephrasing

Pg 6 Last sentence needs data or citation

Pg 7 First paragraph last sentence “became disappeared” - awkward phrasing. Delete became.

Pg 16 Last line “conjecture” - does not fit context, maybe “conjunction” instead

Multiple pages - “even not RGG-/R-rich” - awkward phrasing, perhaps “that are not RGG-/R-rich” if that is the correct meaning.

Response: Thanks so much for the comments and we have corrected them all.

Reviewer 2.

MAJOR CONCERNS:

1. Throughout the manuscript, the authors claim several observations for which they do not provide any experimental data. The authors need to present these data, at least in the supplementary information. Without the experimental data, it's hard to believe their observations/results.

Response: Thanks so much for the kind comments.

Previously, 1) we did not present the DIC images of samples without droplets because all the samples in the form of homologous solutions have nothing in the DIC view field. In our previous papers on LLPS, we once included such images but many reviewers requested to remove them and suggested to only have the description in the text. To address the comment, now in the revised manuscript, we presented the DIC images without LLPS (Figure 1C and 1D).

2) For the images of aggregates, we found that once the samples become quickly aggregated, they looked very similar if imaged by DIC and EM even for different mutants. Furthermore, previously the aggregation of TDP-43 PLD has been exhaustively studied by many groups including us. In fact, my group has been focused on studying aggregation of TDP-43 PLD since 2010 and published the first systematic study on aggregation and amyloid fibrillation of TDP-43 PLD and its ALS-causing mutants (Lim et al., Song [2016] *PLoS Biology*. Ref. 16). As such, in the current study, aggregation is not the focus but only the observation during the experiments such as those for screening the conditions which allowed us to conduct NMR visualization of the molecular interaction of ATP with TDP-43 PLD in LLPS.

3) As my lab started to work on TDP-43 since 2008, before the current study, many other graduate students and postdocs have conducted extensive screening of protein concentrations, buffer conditions suitable for characterizing intrinsically-prone TDP-43 including N-terminal domain (Qin et al. Song [2014] *PNAS*. Ref. 4). Therefore, even for LLPS of TDP-43 PLD, other students have previously studied and published. So the present optimization of NMR conditions was largely based on our previous results. For most cases, once we found some samples got aggregated visibly, we just moved to testing other conditions without detailed characterization of these aggregates by DIC or EM.

To address this comment, we presented some images of aggregates in the revised Figure S1B.

2. Also, a prevalent weakness of the manuscript is a lack of rigor in assessing the phase separation behavior of the multiple TDP-43 PLD constructs that are reported. This is done largely on the basis of non-quantitative analyses of DIC microscopy images. A rigorous approach would be to determine C_{sat} values for the different constructs by varying protein concentration and establishing quantitative image analysis criteria that define when phase separation/aggregation occurs. Additionally, the authors should provide phase diagrams either by turbidity assay or DIC microscopy images for TDP-43 PLD LLPS in the absence and presence of ATP for the readers to appreciate the change in the C_{sat} values.

Response: Thanks so much for the kind suggestions.

Indeed, to establish the complete phase diagrams of TDP-43 WT- and mutant PLD itself represents an important direction for TDP-43 research. In fact, in the field of polymer phase separation, many excellent papers are only focused on mapping

out the phase diagram. However, for protein phase separation, this task becomes extremely challenging and could be done mostly for well-behaved model peptides, or very limited numbers of real proteins.

Unfortunately, particularly for TDP-43 PLD, it is almost impossible to achieve this task because TDP-43 PLD is intrinsically aggregation-prone, as evidenced by the absence of any publications on phase diagram for LLPS of TDP-43 PLD so far. As many groups including us previously observed, only after extensive screening, a very narrow window of protein concentrations and buffer conditions might be found to allow biophysical/NMR studies of TDP-43.

In particular, our current study is not aimed to obtain the quantitative phase diagrams/behaviours of TDP-43 PLD. Instead, the novelty and significance of our current study is to successfully decipher the residue-specific molecular interactions of ATP on modulating LLPS of TDP-43 by residue-specific NMR characterization, which does not require the quantitative phase diagram/behaviours.

Moreover, although NMR spectroscopy is very powerful in identifying atomic-resolution interactions, its requirement for the sample quality/solubility is much more strict than most other biophysical methods. For example, even dynamic association of TDP-43 PLD is sufficient to prevent NMR characterization.

-Following is the list of the author's observations, for which they do not show any experimental data in the manuscript.

1) In the first section of the results (first paragraph, Page 6), "concentrations > 50 μ M, TDP43 PLD phase separates without ATP but undergo aggregation after 1 h".

Response: Thank you so much for this suggestion. We have now presented the DIC images of aggregates in Figure S1.

2) In the first section of the results (second paragraph, Page 6), "DIC images showed that without ATP, no droplets or aggregates were formed".

Response: Thank you so much. We have presented the DIC images of the samples without LLPS in Figure 1.

3) In the first section of the results (second paragraph, Page 6), "the DIC image at 1:1000 showed dissolution of droplets".

Response: Thank you so much. We have presented the DIC images of the samples without LLPS in Figure 1.

4) In the second section of the results (third paragraph, Page 8), "DIC images indicated that no droplets or aggregates were detectable at all points of AMP concentrations".

Response: Thank you so much for this comment. For this sample, the DIC image looked the same as we presented in the revised Figure 1.

5) In the second section of the results (fifth paragraph, Page 8), "DIC images indicated that no droplets or aggregates were formed in the presence of NaCl even up to 150 mM".

Response: Thank you so much for this comment. For this sample, the DIC image looked the same as we presented in the revised Figure 1.

6) In the third section of the results (second paragraph, Page 9), “All of the mutants (C2, C3, and C4) become severely prone to aggregation at pH 6.0”.

Response: Thank you so much for the comment. We now presented the images of aggregates in the revised Figure S1.

7) In the third section of the results (second paragraph, Page 9), “At pH 5.5, the C2 and C3 samples showed no significant aggregation at 15 μ M in 10 mM phosphate buffer.”

Response: Thank you so much for the comment. We now presented the images of aggregates in the revised Figure S1.

8) In the third section of the results (third paragraph, Page 10), “DIC images showed that without ATP, the C2 sample had no droplets or aggregates. Also, droplets got completely dissolved at 1:1000 molar ratio”.

Response: Thank you so much for this comment. For this sample, the DIC image looked the same as we presented in the revised Figure 1.

9) In the third section of the results (fourth paragraph, Page 10), “DIC images indicated that the C3 sample had no droplets or aggregates without ATP. Also, for the C4 sample, the addition of ATP at 1:10 induced precipitation.

Response: Thank you so much for the comment. We now presented the images of aggregates in the revised Figure S1.

10) In the third section of the results (fifth paragraph, Page 10), “Conducted the ATP titrations on three constructs, i.e., C2, C3, and C4. However, the data for the C4 construct is not shown in the manuscript.

Response: Thank you so much for this comment. There were mistakes with numbering of the constructs and now we corrected the mistakes. On the other hand, once a sample got aggregated, its DIC image looked like those in Figure S1 and NMR spectrum showed no signal.

11) In the fourth section of the results (third paragraph, Page 12), “DIC images revealed that for C5, without ATP, no droplets were detected while with the addition of ATP at 1:25, some droplets were formed”.

Response: Thank you so much. The DIC images looked like what were presented in Figure 1.

12) In the fourth section of the results. The authors do not show the DIC images for both the C6 and C7 mutants.

Response: Thank you so much for pointing out the mistakes. There were mistakes with numbering of the constructs and now we corrected the mistakes. On the other hand, once a sample got aggregated, its DIC image looked like those in Figure S1 and NMR spectrum showed no signal. If so LLPS occurred, its DIC image look like those in Figure 1.

MINOR CONCERNS:

1) Figure 1- Panel A-The authors should replace 1, 2, 3, 4, 5, 6, 7 with WT, C2, C3, and so on because the authors have used the latter notations for referring to different constructs in the manuscript. Also, please mention in the figure

caption that the numbers written in blue are the pI values. In Panel B, the authors should mention the wavelength at which the turbidity values were measured (on the y-axis). Also, the colors used are confusing here. I would suggest using different symbol colors and the same line color. This same comment is for Figure S1B as well.

Response: Thank you so much for these kind suggestions.

1) We have now relabelled the constructs as the reviewer kindly suggested. 2) We also added the wavelength (600 nm) into both figure legend and main text. 3) We made all the new turbidity curves as the reviewer kindly suggested.

2) In the introduction (first paragraph, Page 3), "...despite being intrinsically-disordered, TDP-43 PLD host almost all ALS-causing mutations...." And, (second paragraph, Page 3), "...cytoplasmic TDP-43 droplets act to recruit importin- α and Nup62...." The authors need to provide the references here.

Response: Thank you so much for this suggestion and we have added the reference.

3) In the introduction (third paragraph, Page 4), "We further found that by specific binding, ATP also mediates LLPS..." The authors should rephrase this sentence as this observation is from their previous findings.

Response: Thank you so much for this suggestion and we have revised it.

4) In the introduction (fourth paragraph, Page 4), Suggest rephrasing: "For the first time, we found that ATP is indeed capable...."

Response: Thank you so much for this suggestion and we have revised it.

5) In the first section of the results (first paragraph, Page 6), there is a typo (25 °C). The authors should also specify whether they have used NaCl in their studies (in Figure 1, Figure 2, etc.). If yes, then the authors should mention the concentration, either in the figure caption or in the Methods. And, (second paragraph, Page 6). suggest rephrasing: "... 1.62 and 0.94 respectively at pH 7.0 and 5.5".

Response: Thank you so much for this suggestion and we have added them.

6) In the second section of the results (second paragraph, Page 7), "As it is well-documented that the C-terminal...." The authors need to provide the reference here.

Response: Thank you so much for this suggestion and we have revised it.

7) In the second section of the results (fourth paragraph, Page 8), the authors state that the DIC images of TDP-43 PLD in the presence of ADP at 1:200 molar ratio induced the formation of detectable aggregates. However, in the micrographs, aggregates are not visible at this ratio. The authors should comment on this.

Response: Thank you so much for pointing this typo. It should be 1:400 and we have corrected it.

8) The authors report rapid aggregation of a few of the mutant constructs, specifically the ones lacking all of the Arg residues. Do the authors know

whether these aggregates are amorphous or amyloid-like? The authors should comment on this in the manuscript.

Response: Thank you so much for this comment.

As we provided above, the aggregation of TDP-43 PLD has been exhaustively studied by many groups including us before. So here our focus is to delineate the high-resolution interaction of ATP on modulating LLPS. As such, we did not go further to characterize these aggregates as we previously conducted (Lim et al. Song [2016] *PLoS Biology*), which costed a large quantity of funding, manpower and time.

9) The scale bars should be provided for all of the DIC images (for an individual image), and the details can be specified in the figure caption in case the space is less.

Response: Thank you so much for this kind suggestion. For one set of DIC images, we always used the same scale and this allowed us to use one scale bar for one sub-figure. We have tried to place scale bars for all images but found that the scale bars would cover up the droplets.

10) In the figure caption 2C, please mention that the data in the presence ATP is shown in red.

Response: Thank you so much for kindly pointing out this mistake and we have added it.

11) In the figure caption 3D, either write upper & lower or I & II to refer to the TDP-43 C5and C7-PLD.

Response: Thank you so much for kindly pointing out this and we have added them into the legend.

12) In the figure caption S1B, please mention the molar ratio of TDP-43 PLD and ATP/ADP/AMP used for these studies.

Response: Thank you so much for this suggestion and we have added them into the legend.

13) In the figure caption S3B, please specify which data is shown in blue.

Response: Thank you so much for this suggestion and we have added it.

14) Many of the references are erroneously formatted. The authors may want to export article citations directly from PubMed using EndNote or some other reference management software to standardize their format.

Response: Thank you so much for this kind suggestion. The journal appears to allow different formats of references. We will reform them if the journal requests in the future.

Reviewer 3.

1) On page 6, a reference is required “.....On the other hand, at concentrations > 50 μ M, it could phase separate without ATP but would exaggerate into aggregation after 1 hr.”

Response: Thank you so much for this kind suggestion and we have revised and added the references.

2) On page 7, the authors mentioned about “that the number of the droplets was higher than that at pH 5.5 (Fig. 1C), thus resulting in the higher turbidity (Fig. 1B).” I am wondering about the pI of the wild-type TDP-43 PLD protein and if higher pH induces more aggregation, then is there any correlation between the number of droplets and aggregation propensity? What would be the author’s comment about it?

Response: Thank you so much for this kind comment.

As LLPS and aggregation are governed by the interplay of a variety of multivalent and dynamic interactions, it is extremely challenging to establish their high-resolution mechanisms, including the quantitative relationship for LLPS and aggregation of TDP-43 PLD.

Fortunately, the new results with all-Lys mutant allowed us to propose a speculative mechanism which may provide an insight into this relationship. As this mechanism is involved in extensive revisions in results and discussions, it is impossible to copy them here. Please kindly find the discussion in the revised manuscript.

3) In figure 1C, the Image scale is missing. I am assuming that the scales are the same. Then, is the size of the droplets at pH5.5 different from pH 7.0? Did authors see any different droplet fusion at these two pHs?

Response: Thank you so much for the comment.

For one set of DIC images, we always used the same scale and this allowed us to use one scale bar for one sub-figure. We have tried to place scale bars for all images but found that the scale bars would cover up the droplets. The size at two pH have no significant difference and the fusion would occur if the droplets sink to the bottom of the tube after a while.

4) In figure 1D (1H- 15N NMR HSQC spectra), few red peaks looked broader or may be clustered. What do authors think whether it may be for protein aggregation or due to rapid amide proton exchange with solvent or fast protein dynamics at 1:1000 molar ratio?

Response: Thank you so much for this comments.

We have added a sentence to speculate the mechanism underlying this observation as: “Interestingly, HSQC peaks of TDP-43 PLD in the presence of ATP at 1:1000 appeared to be broader than those in the free state, which may be due to the alternation of dynamics, or/and association of PLD upon binding to ATP”.

5) On page 9, probably, the authors wanted to mean C4 instead of C3 in this sentence “.....it is unexpected for the proneness of C3 with pI of ~3.6”.

Response: Thank you so much for pointing out this mistake. As initially we named WT to be C1 but later changed, there existed several mistakes with the numbering. We have now corrected them all.

6) This is a very interesting observation for C4 as they mentioned “Noticeably, even for the transparent C4 sample immediately with aggregates removed by centrifuge, the addition of ATP at 1:10 induced precipitation as detected by DIC. Moreover, the precipitation still occurred even at a protein concentration of 5 μ M upon adding ATP at 1:10.” In the C4 construct, the authors deleted all the key Arg residues. What will be the molecular mechanism of this observation or which interactions are responsible for the formation of this turbidity in the context of these Arg residues?

Response: Thanks so much for this critical comments.

With the new results on all-Lys mutant, we were able to propose a speculative mechanism to address the roles of Arg in maintaining the reversibility of LLPS and preventing aggregation. As this speculation is involved in extensive revisions in results and discussions, please kindly find them in the revised manuscript.

7) The authors did not mention how long did they wait before taking their reading in DIC and NMR? I am wondering about the lifetime of these droplets.

Response: Thank you so much for this suggestion. We waited for 15 min after the sample preparation and now we added it in Methods.

8) In figure 2B, the authors marked C2 and C3 columns in blue as “Nm-Cw and Nw-Cm”. The authors did not mention about them.

Response: Thank you so much for pointing out this mistake. We have removed them.

9) Authors used 5 μ M for the DIC experiment and for most of the NMR experiment, they used 15 μ M. Is there any reason for using two different concentrations for these two experiments?

Response: Thanks so much for pointing out this confusion.

In fact, we used 15 μ M for both DIC and NMR for all constructs except for C5 which is highly aggregation-prone.

10) I am wondering whether authors should give the number (III) image for their speculative model as they only established (I),(II), and (IV) by their experiments. I understood that they tried to cover the bigger picture but, they did not do any experiment with Nup62 and Importin- α .

Response: Thank you so much for this comment.

We would apologize for the confusion. In fact, the whole speculative diagrams from I to IV were built up on the previous studies by other groups. Our current results only provide a biophysical view of these processes. So in the revised manuscript, we have revised and added the references to avoid this confusion.

11) It will be helpful if authors will provide a reference for this “.....it rationalizes the observed susceptibility of the intrinsically-disordered domains to minor perturbations such as the Met337Val mutation of TDP-43 PLD, which is sufficient to significantly alter the properties of its LLPS and aggregation.”

Response: As this point was also commented by the first reviewer. We removed this sentence as we do not have the data on mutants.

12) M414 showed significant chemical Shift difference (CDS) (ppm) for almost all the construct that the authors used. What will be the author's interpretation of the role of M414 in the formation of LLPS? Has M414 any interaction with ATP or is M414 a part of the delicate network in the cytoplasmic TBP-43 sequence?

Response: Thank you so much for this suggestion and we have added it.

13) Following question 6 again, in the manuscript, authors showed that one variant without Arg residues showed a high propensity to aggregates, whereas, other five variants showed the same trend of turbidity at low concentration of ATP and then the disappearance of this LLPS at high concentration of ATP. Thus, the manuscript has a very interesting observation of the role these five Arg residues to form LLPS and then aggregates. Thus, for the LLPS formation, these Arg residues are required but what is about aggregation (like their C4 construct)? In the discussion, an explanation of the connection between LLPS and aggregation in the context of these variants will be more interesting to the readers.

Response: Thank you so much for this kind suggestion.

Fortunately, the new results with all-Lys mutant allowed us to propose a speculative mechanism to address this comment. As this speculation is involved in extensive revisions in results and discussions, please kindly find them in the revised manuscript.

14) I am not aware of the journal reference policy but the authors gave 9 self-citations out of 43 references, which is nearly 21%.

Response: Thank you so much for this comment.

As you could kindly see, so far only my group is using NMR to study the functions of ATP on proteins at mM. So to adequately introduce the background and to support the interpretation of the current results, we have to cite a minimal portion of our papers on this topic.

On the other hand, the previous manuscript was prepared for a journal which has a tight limitation of the reference number. As such, we were also unable to cite many other references which led to the high-percentage of the self-citation. As Communications Biology has no tight limitation of the reference number, we now resolved this problem by citing more papers by other groups on the relevant topics.

REVIEWERS' COMMENTS:

Reviewer #1 (Remarks to the Author):

The addition of the all-Lys variant experiments has greatly enhanced the impact of the work. Performing the NMR-based titration experiments illustrates that ATP binds with less affinity to the all-Lys TDP-43 construct vs. the WT Arg-containing TDP-43. This is an important finding for the LLPS field. Additionally, the effects of the Arg residues on the overall propensity of TDP-43 to phase separate vs. aggregate is a timely contribution, albeit much still remains to be discerned about the complex interplay of intramolecular and intermolecular interactions involving the PLD as well as the hydrophobic domain important for mediating TDP-43 PLD LLPS.

A couple of caveats should be mentioned: The first is that the effects of ATP on NMR spectra are visualized on the amide backbone resonances of Arg and Lys. A sentence should be included to emphasize this point in the discussion. The second is that the study concerns the PLD of TDP-43; it will be very interesting to determine the effects of the N-terminal domain and RRM on the interaction with ATP, but this is beyond the scope of the current work.

As this study illustrates the role of ATP in potentially binding Arg residues, more details in the methods section regarding preparation of ATP solutions for experiments should be included. How much MgCl₂ was added to 'stabilize' the ATP-Mg complex? How were the ATP solutions prepared and checked for changes to pH? At pH 5.5, sodium phosphate is not an optimal buffer so I wonder about the effects of ATP on the overall pH of the protein solutions used here. As more studies continue to focus on the effects of AMP, ADP, and ATP on phase separation, the community should be very cognizant of how solutions are prepared to ensure reproducibility across experiments from different groups.

There are a number of minor corrections to be made in the text and they are listed here:

The last sentence of the introduction is a run-on sentence (7 lines, lines 105-111). Please reword into clear sentences.

Line 166 "Noticeably,"

On Line 167, the authors reference the titration experiment that is more clearly represented in the last figure in the paper (Figure 7A), whereby the authors show that the residue CSPs are saturated by the time the 1:100 ratio is reached.

Line 203: or 'due' to the secondary perturbation

On lines 233-236, I was confused as the authors reference C3, but Figure 3B shows DIC microscopy for C1 and C2? It's important to get this section right, as the C3 construct does not contain any Arg residues.

Line 317: "further depends on the presence"

Line 322: "slightly lower pI"

On Figure 6, it would be very helpful to the reader to show an overlay of the turbidity curves for WT and the all-Lys variant (C7) for visual comparison, particularly as Figure 1 and 6 have different ATP:protein ratios.

Line 418: "coordinate other driving forces"

Line 434: "aggregation of TDP-43 PLD becomes emerging" needs to be reworded.

Line 465: "will become dominant"

Line 468: "ATP can achieve relatively-tightly binding" needs to be reworded.

Reviewer #2 (Remarks to the Author):

The authors have largely addressed the comments of the reviewers and revised the manuscript accordingly. They have now provided the experimental data and detailed explanation, wherever required. From my point of view, the paper is almost suitable for publication in Communications Biology. However, I recommend publication only after the following minor concerns are addressed.

1) The title of the paper needs to be rephrased.

2) I understand that the current study's focus is to decipher the residue-specific molecular interactions of ATP on modulating LLPS of TDP-43 by residue-specific NMR characterization. LLPS, as the authors surely know, is concentration-dependent. Therefore, it's important to report the concentration range where TDP-43 undergoes LLPS and aggregation to better understand the phase behavior of TDP-43. I suggest that the authors should provide concentration-dependent turbidity values or DIC images of TDP-43 LLPS/aggregation in the presence of ATP in the SI and can perform dissolution experiments to distinguish LLPS from aggregation.

3) The authors should provide data in the SI instead of saying that "the DIC image looked the same" since the solution conditions are different in these cases.

Reviewer #3 (Remarks to the Author):

No Comment